# Presentation and Clinical Course of Leptospirosis in a Referral Hospital in Far North Queensland, Tropical Australia

**DOI:** 10.3390/pathogens14070643

**Published:** 2025-06-28

**Authors:** Hayley Stratton, Patrick Rosengren, Toni Kinneally, Laura Prideaux, Simon Smith, Josh Hanson

**Affiliations:** 1Department of Medicine, Cairns Hospital, Cairns, QLD 4870, Australia; hayley.stratton@nt.gov.au (H.S.); toni.kinneally@health.qld.gov.au (T.K.); laura.prideaux@health.qld.gov.au (L.P.); simon.smith2@health.qld.gov.au (S.S.); 2School of Medicine and Dentistry, James Cook University, Cairns, QLD 4870, Australia; patrick.rosengren@my.jcu.edu.au; 3The Kirby Institute, University of New South Wales, Kensington, NSW 2052, Australia

**Keywords:** leptospirosis, clinical management, critical care, tropical medicine, zoonotic infection, tropical Australia

## Abstract

The case-fatality rate of severe leptospirosis can exceed 50%. This retrospective cohort study examined 111 individuals with laboratory-confirmed leptospirosis admitted to Cairns Hospital, a referral hospital in tropical Australia, between January 2015 and June 2024. We examined the patients’ demographic, clinical, laboratory and imaging findings at presentation and then correlated them with the patients’ subsequent clinical course. Severe disease was defined as the presence of pulmonary haemorrhage or a requirement for intensive care unit (ICU) admission. The patients’ median (interquartile range) age was 38 (24–55) years; 85/111 (77%) were transferred from another health facility. Only 13/111 (12%) had any comorbidities. There were 63/111 (57%) with severe disease, including 56/111 (50%) requiring ICU admission. Overall, 56/111 (50%) required vasopressor support, 18/111 (16%) needed renal replacement therapy, 14/111 (13%) required mechanical ventilation and 2/111 (2%) needed extracorporeal membrane oxygenation. Older age—but not comorbidity—was associated with the presence of severe disease. Hypotension, respiratory involvement, renal involvement and myocardial injury—but not liver involvement—frequently heralded a requirement for ICU care. Every patient in the cohort survived to hospital discharge. Leptospirosis can cause multi-organ failure in otherwise well young people in tropical Australia; however, patient outcomes are usually excellent in the country’s well-resourced health system.

## 1. Introduction

Leptospirosis is a life-threatening zoonotic infection caused by pathogenic spirochaetes of the genus *Leptospira* [1]. The incidence of leptospirosis is highest in low- and middle-income countries in tropical and subtropical regions [2]. In 2015, it was estimated that there were over 1 million annual cases of leptospirosis worldwide, with ~60,000 deaths and the loss of 2.9 million disability-adjusted life years [1,2,3]. With urban expansion and, potentially, the impact of anthropogenic climate change, the global incidence of leptospirosis is expected to rise [4,5].

Mammals, the primary reservoir for *Leptospira* organisms, pass urine into the environment where the organism can survive for many months in soil and fresh water [1]. Humans become infected by direct contact with infected animals or after exposure to the *Leptospira* organism in the environment through a breach in the skin or penetration of mucous membranes or the conjunctiva [1]. Infection commonly occurs after occupational or recreational exposure, although the incidence of leptospirosis is also higher in people living in flood-prone, overcrowded urban areas with large rodent populations [6,7,8,9]. The incidence of leptospirosis is highest during warmer months and during periods of heavy rain, which both favour the survival of *Leptospira* organisms in the environment [10,11].

The clinical presentation of leptospirosis can range from mild, non-specific symptoms that include fever, headache and myalgia, to life-threatening manifestations that include shock, pulmonary haemorrhage, acute kidney injury and multiorgan failure requiring intensive care unit (ICU) admission [1,12]. The case fatality rate of severe leptospirosis can exceed 50% but reported rates vary enormously and this variation is likely to be explained by geographical differences in the resourcing of health systems, which particularly impacts access to diagnostic laboratory services and to advanced supportive care [1,13,14,15,16,17].

Far North Queensland (FNQ), a region of tropical Australia, has the highest incidence of leptospirosis in the country [18,19]. In approximately 85% of local cases of leptospirosis, there is a history of potential occupational or recreational exposure to the pathogen, with almost 90% of the cases occurring in districts where there is high-intensity banana and cattle farming [20,21]. FNQ’s largest hospital—Cairns Hospital—has the region’s only ICU. About 11% of laboratory-confirmed adult cases in FNQ require admission to the Cairns Hospital ICU, but the case fatality rate of patients requiring ICU admission at Cairns Hospital between 1998 and 2018 was only 4% [12]. This high survival rate can be explained by local familiarity with the disease, a well-resourced retrieval network and access to advanced multimodal critical care support in Australia’s universal health system [12,22]. Recently, there have been further enhancements in local critical care services, which have included the establishment of a statewide extracorporeal membrane oxygenation (ECMO) retrieval service, the employment of more ICU specialists and the expansion of high dependency unit services at Cairns Hospital.

This study was performed to further define the clinical phenotype of laboratory-confirmed leptospirosis in the FNQ region of tropical Australia. The study also examined the epidemiological, clinical and laboratory characteristics of the patients who developed severe disease. It was hoped that these data would inform the optimal contemporary care of individuals with leptospirosis in the region.

## 2. Methods

This retrospective study was performed at Cairns Hospital, the 771-bed referral hospital for the FNQ region. The hospital serves a population of ~290,000 people who live across an area of 380,000 km^2^ in Northeast tropical Australia, between the latitudes of 9.27° S and 18.26° S. The region has a tropical climate, with hot, humid summers and milder, drier winters. The average annual rainfall is 1992 millimetres. Most cases of leptospirosis present during the region’s December–April wet season, when the monthly rainfall is often greater than 400 mm [10].

The FNQ region contains two health districts: the Cairns and Hinterland Hospital and Health Service (CHHHS) and the Torres and Cape Hospital and Health Service (TCHHS). The CHHHS serves the population of Cairns (FNQ’s main city) and the surrounding regions, while the TCHHS serves the remote population of the Cape York Peninsula and Torres Strait Islands (Figure 1).

Only individuals who were admitted to Cairns Hospital between 1 January 2015 and 30 June 2024 with a laboratory-confirmed diagnosis of leptospirosis were eligible for this study. The study period was chosen as it coincided with the local introduction of an electronic medical record (EMR) system. Potential study participants were identified by interrogating the Queensland public health system’s electronic laboratory database (AUSLAB). All participants had to satisfy the Australian definition of laboratory confirmed leptospirosis, meeting one or more of the following four criteria: (1) isolation of pathogenic *Leptospira* species from blood culture; (2) a single *Leptospira* microscopic agglutination titre (MAT) ≥ 400 supported by a positive enzyme-linked immunosorbent assay IgM result; (3) a fourfold or greater rise in *Leptospira* microscopic agglutination titre between acute and convalescent phase sera obtained at least two weeks apart or (4) detection of *Leptospira* in blood by polymerase chain reaction (PCR) [24]. Culture of *Leptospira*, PCR testing (target outer membrane protein *LipL*32) and MAT testing were performed at Australia’s Leptospirosis Reference Laboratory in Brisbane, 1390 km away from Cairns Hospital. The panel of serovars that were tested is presented in Appendix A. Urinary culture is not performed in Queensland due to low yields and high rates of contamination. The use of urinary PCR was not validated for clinical use at the reference laboratory during the study period.

The patients’ EMR was reviewed to collect demographic data and to identify any comorbidities (Appendix A). Individuals living in the TCHHS were said to have a remote residence, while the individuals living in the CHHHS, but outside the city of Cairns, were defined as having a rural residence. The patients’ symptoms and clinical signs that were documented at their presentation to Cairns Hospital were also recorded. The patients’ SPiRO score, a marker of disease severity, was calculated as defined previously [21]. Laboratory data were collected from AUSLAB; these data included the values at presentation (either at the regional referring hospital or Cairns Hospital) and the highest or, where relevant, lowest values during the patient’s hospitalisation at Cairns Hospital. A patient was defined as having acute kidney injury at presentation if their estimated glomerular filtration rate was <60 mL/min/m^2^; they were said to have acute liver injury at presentation if they had a serum alanine transferase greater than five times the upper limit of normal (≥150 IU/mL) or a total serum bilirubin greater than three times the upper limit of normal (≥60 µmol/L). Oliguria at presentation was recorded if it was documented in the medical record. All diagnostic testing for leptospirosis—including PCR, serology, and culture—was sought, and the results were recorded. Specialist radiologist and cardiologist reporting of any imaging was also collated.

The patients’ treatment and their clinical course were also reviewed. There was a particular focus on the antibiotics that were prescribed, the time to antibiotic therapy and the duration of antibiotic therapy. If patients developed pulmonary haemorrhage, required ICU admission, required vasopressor support, required intubation and mechanical ventilation, required renal replacement therapy (RRT) or required transfer to another facility for ECMO, this was recorded. Pulmonary haemorrhage was said to be present if there was documented frank haemoptysis or if frank blood was present on tracheal aspirate. Individuals were defined as having severe leptospirosis if they were admitted to the ICU, or they had pulmonary haemorrhage. The patients’ in-hospital and 90-day mortality were recorded. The patients’ length of stay in hospital and, if relevant, the ICU was also documented.

### 2.1. Statistical Analysis

Data were de-identified, entered into an electronic database (Microsoft Excel, version 16.0) and analysed with statistical software (Stata version 18.0). Groups were analysed using the Wilcoxon rank-sum test, the chi-squared test, Fisher’s exact test or logistic regression, where appropriate. Trends over time were analysed using Spearman’s test for correlation. If individuals were missing data, they were not included in analyses which evaluated those variables.

### 2.2. Ethical Approval

The study was conducted in accordance with the Declaration of Helsinki and approved by the Far North Queensland Human Research Ethics Committee (HREC/EX/2024/QCH/108994) on 2 August 2024. As the retrospective data were de-identified and presented in an aggregated manner, the Committee waived the requirement for informed consent.

## 3. Results

There were 111 individuals who satisfied the inclusion criteria for the study; 94/111 (85%) were male, and 86/111 (77%) were transferred to Cairns Hospital from another health facility within the CHHHS or TCHHS. The median (IQR) age of the cohort was 38 (24–55) years. The cohort included six children; just one of these children was younger than 10 years of age (Table 1).

Most individuals (67/111, 60%) presented during the local December–April wet season (Appendix A). In 81/111 (73%), there was a potential occupational or environmental exposure preceding the patient’s presentation; this was most commonly farming or recreational exposure to freshwater (Appendix A). The number of cases admitted to Cairns Hospital increased between 2015 and 2023, the first and last complete years in the study period (r_s_ = 0.75, *p* = 0.02) (Figure 2).

### 3.1. Disease Severity and Clinical Course

All 111 individuals survived to hospital discharge, and all were alive 90 days after their presentation. The median duration of hospitalisation was 5 days with an interquartile range (IQR) of 3–8 days. There were 63/111 (57%) in the cohort who satisfied criteria for severe disease. This included 56/111 (50%) who required admission to ICU; the median (IQR) duration of their ICU stay was 3 (2–5) days. There were 4/111 (4%) who were transferred from Cairns Hospital to another centre for escalation of care (including ECMO) or, as Cairns Hospital does not have a paediatric ICU, because of their age.

### 3.2. Correlation Between Age, Comorbidity and Subsequent Clinical Course

Severe disease was more common in older patients, but 40/63 (63%) of patients with severe disease were younger than 50 years of age (Appendix A). There were 13/111 (12%) individuals in the cohort who had a documented comorbidity. There was no statistically significant association between the presence of comorbidity and severe disease. Indeed, 53/63 (84%) with severe disease had no comorbidity (Table 1).

### 3.3. Diagnosis of Leptospirosis

PCR was performed on blood in 99/111 (89%) and was positive in 89/99 (90%). Whole-blood culture was performed in 48/111 (43%) and was positive in 30/48 (63%). Acute serology testing was performed in 104/111 (94%) and was positive in 45/104 (43%). Convalescent serology was collected and tested in 51/111 (47%). This was frequently used to define the infecting serovar, but in 12/51 (24%), it established the diagnosis of leptospirosis. It was possible to determine the serovar in 59 individuals; the most commonly identified serovars were *Leptospira interrogans* serogroup Pyrogenes serovar Zanoni (21/59, 36%) and *L. interrogans* serogroup Australis serovar Australis (12/59, 21%) (Appendix A). A greater proportion of individuals diagnosed with infection with the Zanoni serovar had severe disease than individuals diagnosed with infection with other serovars, but the difference failed to reach statistical significance in this modestly sized cohort (Table 1). In 71/111 (64%) presentations, leptospirosis was in the admitting clinician’s differential diagnosis.

### 3.4. Clinical Signs and Symptoms at Presentation and Correlation with Subsequent Clinical Course

The median (IQR) duration of symptoms in patients with severe disease was 5 (3–6) days compared to a duration of 4 (2–5) days in individuals without severe disease (*p* = 0.07). The most common symptoms on presentation were subjective fevers, myalgia, headache, nausea and vomiting (Table 2). It was notable that no more than 10/26 (38%) individuals presenting directly to Cairns Hospital were febrile (temperature ≥ 38 °C) and that a temperature ≥ 38 °C had been documented previously in just 51/85 (60%) individuals who were transferred to Cairns Hospital from another facility. Conjunctival suffusion was documented at presentation in just 23/111 (21%). Patients with subjective dyspnoea at presentation were more likely to develop severe disease. Patients with hypotension, tachycardia, oliguria, tachypnoea and a requirement for oxygen supplementation on their initial assessment at Cairns Hospital were also more likely to develop severe disease. The sole physical examination finding that was associated with severe disease was abnormal chest auscultation (Table 2). The 3-point SPiRO score calculated on admission to Cairns Hospital was associated with subsequent development of severe disease; severe disease developed in 10/32 (31%) with a score of 0, 16/36 (44%) with a score of 1, 21/26 (81%) with a score of 2 and 16/17 (94%) with a score of 3, *p* < 0.0001 (Figure 3). 

### 3.5. Laboratory Values on Admission and Correlation with Subsequent Clinical Course

The laboratory values on admission, their evolution during the patients’ hospitalisation, and their association with severe disease are presented in Table 3 and Table 4. Acute kidney injury was present in 48/111 (43%) at presentation. Severe disease was more likely to develop in patients with acute kidney injury at presentation (33/48, 69%) than in patients without acute kidney injury at presentation (30/63, 48%); OR (95% CI): 2.42 (1.10–5.31), *p* = 0.03). Acute liver injury was documented in just 23/111 (21%) at presentation, and severe disease was no more likely to develop in patients presenting with acute liver injury (12/23, 52%) than in patients presenting without acute liver injury (51/88, 58%); OR (95% CI): 0.79 (0.32–1.99), *p* = 0.62). There were 25/111 (23%) with an elevated troponin I at presentation; severe disease was more likely to develop in patients presenting with an elevated troponin I (23/25, 92%) than in patients presenting without an elevated troponin I (40/86, 47%); OR (95% CI): 13.23 (2.93–59.61), *p* < 0.0001). There were 105/111 (95%) with urine microscopy available; 28/105 (27%) had significant haematuria (>50 red blood cells × 10^6^/L) and 58/105 (55%) had significant pyuria (>50 white blood cells × 10^6^/L), but neither haematuria nor pyuria had any association with the development of severe disease.

### 3.6. Chest Imaging Findings on Presentation and During Admission

There were 109/111 (98%) individuals who had chest imaging at presentation; it was abnormal in 37/109 (34%). Individuals with abnormal chest imaging at presentation (27/37, 73%) were more likely to develop severe disease during their hospitalisation than those with normal chest imaging (36/72, 50%); OR (95% CI): 2.70 (1.14–6.38), *p* = 0.02. There were 71/109 who had further chest imaging during their hospitalisation, including 45 patients who had normal imaging at presentation. Abnormalities had developed on this chest imaging in 26/45 (58%) who had normal imaging at presentation. The changes on chest imaging were frequently multilobar and alveolar; patients with multilobar and alveolar changes on chest imaging were more likely to have severe disease (Table 5).

### 3.7. Echocardiography

Echocardiography was performed in 25/111 (23%); it was abnormal in 12, 7/12 (58%) had impaired left ventricular function, 2/12 had impaired right ventricular function, and 3/12 had a pericardial effusion.

### 3.8. Antibiotic Therapy

Antibiotic therapy was prescribed to 109/111 (98%); two children (aged 7 and 11, respectively) were admitted to the general ward with fever and observed without antibiotic therapy. By the time leptospirosis was confirmed, their symptoms had abated, and so no antibiotics were commenced. The median (IQR) time from the onset of symptoms to the receipt of antibiotic therapy in the other 109 individuals was 4 (3–5) days. The duration of symptoms prior to receiving antibiotics was longer in patients with severe disease than the duration of symptoms prior to receiving antibiotics in those without severe disease (median (IQR): 5 (3–6) versus 4 (2–5) days, *p* = 0.02).

The patients received a range of antibiotics for a variety of durations, but doxycycline (83/109 (76%)) and ceftriaxone (75/109 (69%)) were the agents that were prescribed most commonly. Broad-spectrum antibiotics (meropenem and piperacillin/tazobactam) were prescribed to 42/109 (39%) for at least some time. Patients admitted to ICU were more likely to receive broad-spectrum antibiotics than those managed outside the ICU (odds ratio (OR) (95% confidence interval (CI)): 9.08 (3.61–22.82), *p* < 0.001). It was possible to determine the duration of antibiotic therapy in 106/109 (97%) individuals who received antibiotics; the median (IQR) duration of antibiotic therapy was 7 (7–10) days. There was one documented Jarisch–Herxheimer reaction; this reaction occurred in a peripheral hospital, before the patient’s transfer to Cairns Hospital.

### 3.9. ICU Care

Of the patients admitted to ICU, 38/56 (68%) needed vasopressor support, 14/56 (25%) required intubation and mechanical ventilation and 13/56 (23%) received RRT. Vasopressor support was, almost always, initially with noradrenaline. In patients requiring mechanical ventilation, lung protective ventilation strategies were employed. In patients receiving RRT, conventional indications for dialysis were employed. There were two patients who required ECMO; both of these patients also needed vasopressor support and RRT. There were six individuals admitted to ICU who did not require vasopressor support, RRT or intubation and mechanical ventilation, but who had critical illness which necessitated close monitoring. Pulmonary haemorrhage was present in 19/56 (34%) who were admitted to ICU and an additional 7 individuals who were not admitted to ICU. There were an additional four individuals who needed vasopressor support and another three individuals who received RRT who were managed without being admitted to the ICU (Figure 4). No patients had plasmapheresis.

### 3.10. Other Management

The patients, in general, received cautious fluid resuscitation. Patients with hypoxia had early use of high-flow oxygen via nasal prongs. A conservative transfusion strategy was employed (with a haemoglobin < 70 g/L as a trigger for blood transfusion and major bleeding with a platelet count < 50 × 10^9^/L as a trigger for platelet transfusion). Patients did not receive corticosteroid therapy routinely, although some patients received corticosteroids as part of their critical care management, particularly those individuals requiring vasopressor support for hypotension. Deep venous thrombosis prophylaxis was pharmacological, although it was generally avoided when the platelet count was <50 × 10^9^/L, when mechanical prophylaxis was preferred. Patients admitted to ICU routinely received proton pump inhibitor therapy for stress ulcer prophylaxis and early enteral nutritional support when indicated.

## 4. Discussion

Every patient admitted to our referral hospital with laboratory-confirmed leptospirosis survived their hospital admission and was alive 90 days later. This is despite the fact that over half of the cohort required ICU admission for supportive care, which included vasopressors, mechanical ventilation, RRT and ECMO. This survival rate compares favourably with series from other parts of the world; indeed the case-fatality rate of severe leptospirosis in one Indian study exceeded 50% [1,13,25]. It is also important to highlight that more than three-quarters of our cohort were transferred from another health facility, which in some cases was over 800 km away. These encouraging outcomes again highlight the success of the hub and spoke model of healthcare in remote, tropical Australia [22].

The explanation for this excellent survival rate is likely to be multifactorial, but it is notable that all but two of the cohort (both children without severe disease) received antibiotic therapy. There is still uncertainty about the benefit of antibiotic therapy in cases of leptospirosis [1,26]. A systematic review in 2024 suggested a shorter time to defervescence, but no effect on mortality or length of hospital stay [27]. A 2024 Cochrane review also highlighted the uncertainty about the ability of antibiotics to reduce mortality or morbidity in individuals with leptospirosis [28]. However, both of these reviews emphasised the paucity of quality studies to examine the issue. While antibiotic therapy was only one element in a suite of interventions for the patients in our cohort, those receiving antibiotics earlier in their disease were less likely to develop severe disease.

There are few downsides to antibiotic therapy in critically ill patients with leptospirosis, as the organism remains sensitive to antibiotics used in empirical regimens for sepsis, including penicillins, cephalosporins and macrolides [1,29]. Although the Jarisch–Herxheimer reaction can occur, it can usually be managed easily with simple supportive care [30]. The laboratory confirmation of the diagnosis of leptospirosis is frequently delayed, and it is important to highlight that the early prescription of antibiotics has been linked to better outcomes in patients with other pathogens, including rickettsial infections and Q fever, which are often difficult to distinguish from leptospirosis [31,32,33,34,35]. Early antibiotic therapy is also a key tenet of recommendations for managing the critically ill patient with other infections [36]. Until data emerge to suggest that the harms of antibiotic therapy outweigh its benefits, we would argue that pragmatic clinicians, particularly those in remote or resource-limited settings with limited critical care support, should continue to prescribe antibiotic therapy for patients presenting with a diagnosis of possible leptospirosis.

The cohort’s excellent outcomes are also likely to be explained by early recognition of high-risk patients at referral sites and enhanced acute care, enabled by a greater awareness of sepsis, the electronic promulgation of clinical guidelines and an effective aeromedical retrieval network in Australia’s well-resourced universal health system [37,38,39]. Once at the referral centre, the patients were also able to receive sophisticated multimodal ICU care including vasopressor support, mechanical ventilation, RRT and, in two cases, ECMO. Patients did not receive corticosteroids routinely, although many received these agents as part of their critical care, particularly in the setting of hypotension unresponsive to fluid resuscitation [40]. Although there are some data to support the use of corticosteroid therapy in individuals with leptospirosis, particularly in those with pulmonary involvement, the well-recognised side effects of corticosteroid therapy argue against their routine prescription in leptospirosis until more definitive supportive data become available [41,42,43]. Plasmapheresis is available in the FNQ region, but it was not prescribed to anyone in the cohort [44].

Almost 70% were younger than 50 years of age, which is likely to be explained by the greater likelihood of occupational and recreational exposure to the pathogen in these populations (Appendix A). The FNQ region has a thriving agricultural sector, and many of the affected individuals are likely to have encountered the pathogen in the course of their work [20]. Exposure was also thought likely to have occurred in several cases during recreational activities that included swimming in freshwater and camping. However, despite the prominence of potential recreational exposure in many of the cases, it was notable that there was only one child under the age of 10 in the cohort. It is also notable that she recovered completely without antibiotic therapy. A lower incidence of leptospirosis in children—and less severe disease—has been noted in other geographic locations [45,46]. Other zoonotic infections in the region—including rickettsial diseases and Q fever—are also less common—and typically cause less severe disease—in children [32,47]. These findings may assist local clinicians in determining the likelihood of these diagnoses while the results of definitive laboratory testing are awaited. Older age was associated with severe disease in this cohort and has been associated with mortality in other studies [2,25,48]. However, more than half of the patients in the cohort with severe disease were younger than 50 years of age without comorbidity, emphasising that leptospirosis can cause life-threatening disease in young, otherwise well individuals.

Leptospirosis was in the initial differential diagnosis of almost two-thirds of the cohort; however, it was notable that some of the “classic” findings of leptospirosis were not apparent. Conjunctival suffusion was documented at presentation in only 20%, a prior fever had been documented in only 63% and only 7% had a bilirubin > 50 µmol/L. This latter point is important because anicteric leptospirosis is often described in the literature as a milder condition [1,25]. However, in our cohort, over half of the patients with a bilirubin < 50 µmol/L at presentation were subsequently admitted to ICU. The lack of association between acute liver injury and severe disease in our cohort is likely to be explained, at least partially, by differences in the locally prevalent serovars [49,50,51]. Jaundice is more likely to be seen in patients infected with the Icterohaemorrhagiae serogroup, a serogroup that is rare in FNQ, where instead serovar Zanoni (Pyrogenes serogroup) and Australis (Australis serogroup) are the most commonly seen (Appendix A) [52]. It is possible that the other clinical manifestations of leptospirosis seen in our cohort—including the absence of fever in many cases and the prominence of hypotension—might relate to differences in the locally prevalent serovars.

Earlier identification of individuals with leptospirosis who are at risk of developing severe disease enables escalation of care. Globally, unfortunately, most cases of leptospirosis occur in socioeconomically disadvantaged individuals living in resource-constrained settings, where access to laboratory support is often limited [1]. The three-point SPiRO score is an entirely clinical score that rapidly identifies high-risk individuals using the three variables of hypotension (SBP < 100 mmHg), oliguria and abnormal findings on chest auscultation correlated with disease severity in this cohort. Our study demonstrates the utility of a simple, but thorough, bedside clinical assessment that identifies high-risk patients who are also likely to require an escalation of care. This approach is equally applicable in other conditions that may resemble leptospirosis in endemic regions at rural or remote sites, where these conditions are frequently more common and where diagnostic support may be limited [53,54,55,56]. Our study—and others to examine the issue—show that basic laboratory tests and simple imaging may refine this assessment further [57,58].

The study again highlights the value of PCR in the prompt diagnosis of leptospirosis [59]. PCR testing was positive in almost 90% of our cohort in whom it was requested early in the patient’s hospitalisation, compared with a figure of 43% for serology. Serological methods also have a lower specificity in a region where leptospirosis is endemic. Serological tests will also have variability in sensitivity and specificity across countries or regions due to differences in the locally prevalent serovars [1]. However, even in Australia’s well-resourced system, access to PCR results was often delayed because testing was performed by the reference laboratory, which is 1390 km away. The development of local PCR diagnostic capacity is a current focus of Cairns Hospital’s laboratory.

Although this study was able to examine the demographic, clinical, laboratory and radiological findings of patients with confirmed leptospirosis in some detail, it has several limitations. The study’s retrospective nature precluded the collection of comprehensive data in all cases. Investigations and clinical assessment were not standardised, and the patients’ symptoms and signs had to be actively sought by attending clinicians and then documented in the medical record. The vital signs that were recorded, and the clinical findings that were documented, were based on the initial assessment in Cairns Hospital, not the findings at the patient’s initial presentation to the health system. The incidence of the Jarisch–Herxheimer reaction is therefore likely to be underestimated as most individuals received their first dose of antibiotics prior to their transfer to Cairns Hospital. It is possible that we have underestimated the contribution of comorbidity to the development of severe disease. Individuals were only said to have a comorbidity if it was documented in the medical record, and it is possible that comorbidities were not documented or had not yet been identified. It is also possible that some of the laboratory indices that were abnormal at presentation were related to unrecognised comorbidity. The referral hospital setting of this study will tend to underestimate the local incidence of the disease while tending to overestimate the local frequency of severe disease. The cohort was heterogeneous and included both adults and children who had a variety of clinical manifestations; there are significant differences in the assessment and management of a child with a non-localising fever for investigation and an older adult with multiorgan failure, although this heterogeneity highlights the protean manifestations of leptospirosis [1]. There is significant geographical variation in the prevalence of different serovars of leptospirosis, which influences the clinical presentation of the cases, limiting the applicability of our findings to other regions [49]. The patients were managed in Australia’s well-resourced universal health system and may therefore be less generalisable to resource-limited settings, where most cases of leptospirosis are seen. However, a high index of suspicion for leptospirosis in the appropriate clinical context, thorough clinical assessment, cautious fluid resuscitation and early referral of high-risk patients to centres where sophisticated critical care support is available are almost certainly likely to be equally relevant in these locations.

Although we understand how the pathogen infects humans, the fact that the number of cases of leptospirosis admitted to this referral centre increased during the study period emphasises the challenges of preventing this life-threatening infection. There is currently no safe and effective universal vaccine to prevent leptospirosis in humans and no conclusive evidence that chemoprophylaxis is effective [1,60]. Although every patient in this cohort survived their hospitalisation, they received a suite of interventions, and it is uncertain which of these therapies provided the greatest benefit. There are major gaps in the understanding of the pathophysiology of leptospirosis, and while host, pathogen and health system factors all contribute to the disease course and outcomes, their relative contributions are incompletely defined [1,50,61,62,63]. From a therapeutic perspective, the incremental value of antibiotic therapy and corticosteroids (in the critically ill patient) has not been established [1]. Future prospective studies must address these issues. We are currently examining, in detail, the presentation and management of the patients in this cohort with cardiac involvement, respiratory involvement and a requirement for ICU admission in an effort to define these clinical phenotypes in more detail and the therapeutic strategies that were associated with their positive outcomes.

## 5. Conclusions

Life-threatening leptospirosis occurs in young people without comorbidity in tropical Australia. Individuals presenting with hypotension, acute kidney injury and evidence of respiratory involvement or myocardial injury are at greatest risk of requiring ICU support. Outcomes are usually excellent as the patients are able to access prompt, multimodal critical care support. The challenge now is to translate these strategies into the resource-limited settings that are disproportionately affected by leptospirosis, but where there is often less access to the prompt diagnostic assessment, the early antibiotic therapy and the sophisticated supportive care that is available in Australia’s well-resourced healthcare system [64,65].

## Figures and Tables

**Figure 1 pathogens-14-00643-f001:**
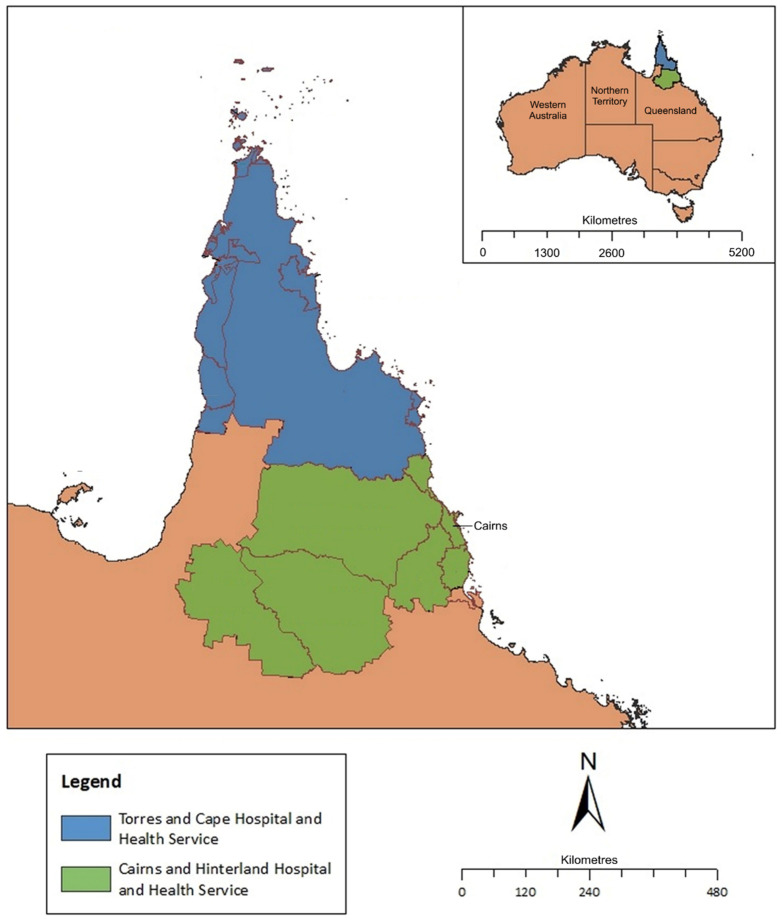
Map of Far North Queensland, Australia, showing catchment area for the current study. Image adapted from Bird, K. et al. [23].

**Figure 2 pathogens-14-00643-f002:**
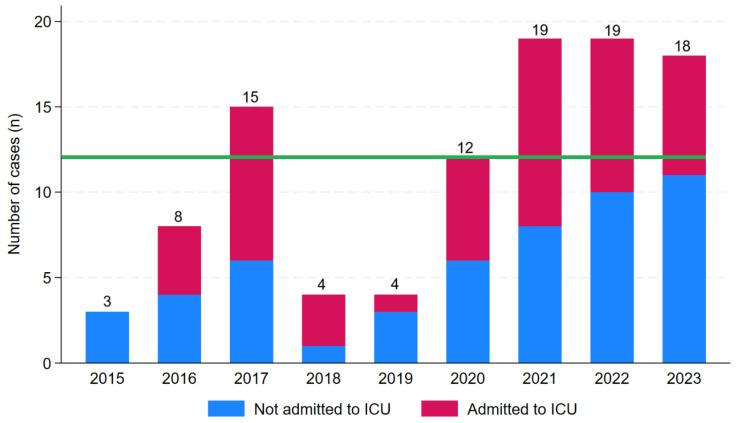
Number of cases of laboratory-confirmed leptospirosis admitted to a referral hospital in Far North Queensland, January 2015 to December 2023 (green line indicates the median annual number of cases). Data from January to June 2024 are not presented. During these 6 months, there were 9 cases, 6 of which were admitted to ICU.

**Figure 3 pathogens-14-00643-f003:**
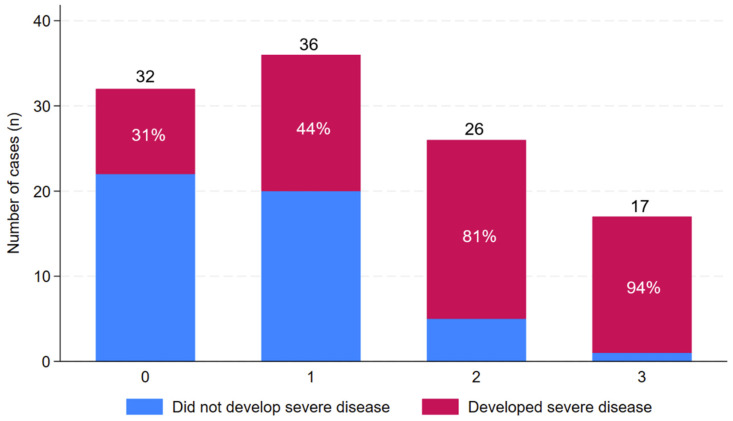
Association between the SPiRO score in patients with laboratory-confirmed leptospirosis at presentation to a referral hospital in Far North Queensland, January 2015 to June 2024, and the development of severe disease [21]. Black numbers above the bars represent the total number of cases. White percentages within the bars report the proportion of these cases that had severe disease (defined as ICU admission or pulmonary haemorrhage).

**Figure 4 pathogens-14-00643-f004:**
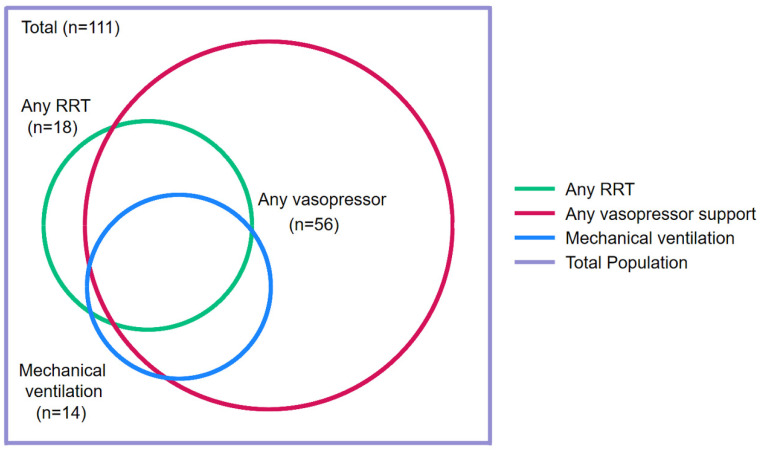
Venn diagram demonstrating the supportive care provided to the patients with laboratory-confirmed leptospirosis in a referral hospital in Far North Queensland, January 2015 to June 2024. RRT: renal replacement therapy. There were 9 patients who received vasopressor support, RRT and mechanical ventilation; 6 patients who received vasopressor support and RRT; 5 patients who received vasopressor support and mechanical ventilation; 36 patients who received only vasopressor support and 3 patients who received only RRT.

**Table 1 pathogens-14-00643-t001:** The demographics and comorbidities of individuals admitted to a referral hospital with leptospirosis in Far North Queensland, January 2015–June 2024, and the association of these factors with the development of severe disease.

Variable	All n = 111	No Severe Disease ^a^ n = 48	Severe Disease ^a^ n = 63	*p*
**Age (years)**	**38 (24–55)**	**32 (19–48)**	**41 (26–63)**	**0.03**
Child (age < 16 years)	6 (5)	3 (6)	3 (5)	1.00
Male sex	94 (85)	38 (79)	56 (89)	0.19
Remote residence ^b^	17 (15)	6 (12)	11 (18)	0.60
Rural or remote residence ^c^	89 (80)	36 (75)	53 (84)	0.23
Wet season presentation ^d^	67 (60)	31 (63)	36 (58)	0.44
Days of symptoms before presentation	4 (2–5)	4 (2–5)	5 (3–6)	0.07
Any comorbidity ^e^	13 (12)	3 (6)	10 (16)	0.15
Diabetes mellitus ^e^	2 (2)	1 (2)	1 (2)	1.0
Cardiac failure ^e^	3 (3)	0	3 (5)	0.26
Ischaemic heart disease ^e^	2 (2)	0	2 (3)	0.51
Chronic kidney disease ^e^	0	0	0	-
Lung disease ^e^	5 (5)	1 (2)	4 (6)	0.39
Liver disease ^e^	5 (5)	1 (2)	4 (6)	0.39
Malignancy ^e^	2 (2)	0	2 (3)	0.51
Autoimmune disease ^e^	0	0	0	-
Immunosuppressed ^e^	0	0	0	-
Hazardous Alcohol use ^e^	29 (26)	12 (24)	17 (27)	0.81
Smoker ^e^	41 (37)	14 (29)	27 (43)	0.14
Serovar Zanoni	21/59 (37)	7/30 (23)	14/29 (48)	0.06
Serovar Australis	12/59 (20)	6/30 (20)	6/29 (21)	0.72

The median (interquartile range) or the absolute number (percentage) is presented. Bold text signifies a statistically significant association. ^a^ Severe disease is defined as ICU admission or pulmonary haemorrhage. ^b^ Residing in the Torres and Cape Hospital and Health Service (TCHHS). ^c^ Residing outside the city of Cairns. ^d^ The local wet season runs from December to April inclusive. ^e^ As defined in Appendix A.

**Table 2 pathogens-14-00643-t002:** The signs and symptoms of individuals admitted to a referral hospital with leptospirosis in Far North Queensland, January 2015–June 2024, and the association of these signs and symptoms with the development of severe disease.

Variable	Number with Data	All n = 111	No Severe Disease ^a^ n = 48	Severe Disease ^a^ n = 63	*p*
**Subjective symptoms**					
**Headache**	**111**	**80 (72)**	40 (83)	**40 (63)**	**0.02**
Fevers	111	106 (96)	45 (94)	61 (97)	0.65
Rigors	111	40 (36)	18 (38)	22 (35)	0.78
Confusion	111	8 (7)	3 (6)	5 (8)	1.00
Fatigue	111	43 (39)	14 (29)	29 (47)	0.07
Abdominal pain	111	42 (38)	20 (41)	22 (35)	0.47
Myalgia	111	83 (75)	34 (71)	49 (79)	0.40
Arthralgia	111	48 (43)	17 (35)	31 (49)	0.15
Diarrhoea	111	41 (37)	14 (29)	27 (43)	0.14
Nausea/vomiting	111	74 (67)	35 (73)	39 (62)	0.14
Chest pain	111	9 (8)	4 (8)	5 (8)	1.00
**Dyspnoea**	**111**	**16 (14)**	**2 (4)**	**14 (22)**	**0.01**
Cough	111	33 (30)	11 (23)	22 (35)	0.17
URTI symptoms	111	15 (14)	6 (12)	9 (15)	1.00
**Haemoptysis**	**111**	**12 (11)**	**0**	**12 (19)**	**0.001**
Abnormal bleeding or bruising	111	11 (10)	3 (6)	8 (13)	0.34
**Objective examination findings**					
Hepatomegaly	111	11 (10)	4 (8)	7 (11)	0.75
Splenomegaly	111	0	-	-	-
Lymphadenopathy	111	6 (5)	1 (2)	5 (8)	0.23
Conjunctival suffusion	111	23 (21)	11 (23)	12 (19)	0.62
Skin rash	111	19 (17)	11 (23)	8 (13)	0.21
**Abnormal chest auscultation**	**111**	**44 (40)**	**13 (27)**	**31 (49)**	**0.02**
**Vital signs at presentation**					
**Oliguria ^b^**	**111**	**42 (38)**	**9 (19)**	**33 (52)**	**<0.001**
Fever > 38.0 °C	111	21 (38)	7 (15)	14 (22)	0.34
**Supplemental oxygen administered**	**111**	**23 (21)**	**2 (4)**	**21 (33)**	**<0.001**
**Respiratory rate ≥ 22 breaths/minute**	**111**	**44 (40)**	**14 (29)**	**30 (48)**	**0.049**
**Heart rate ≥ 100 beats/min**	**111**	**54 (49)**	**17 (35)**	**37 (59)**	**0.02**
**Systolic blood pressure <100 mmHg**	**111**	**56 (50)**	**12 (25)**	**44 (70)**	**<0.001**
**Disease severity score**					
**SPiRO score ^c^**	**111**	**1 (0–2)**	**1 (0–1)**	**2 (1–3)**	**<0.001**

The absolute number (percentage) is presented. URTI: upper respiratory tract infection. Bold text signifies a statistically significant association. ^a^ Severe disease is defined as ICU admission or pulmonary haemorrhage. ^b^ Documented oliguria in notes by the ICU or admitting team. ^c^ Three-point SPiRO score (Systolic blood Pressure < 100 mmHg, Respiratory auscultation abnormalities, Oliguria), each is awarded one point.

**Table 3 pathogens-14-00643-t003:** Haematology values in individuals admitted to a referral hospital with leptospirosis in Far North Queensland, January 2015–June 2024, and the association of these values with the development of severe disease.

Variable	Reference Range ^a^	Number with Data	All n = 111	No Severe Disease ^b^ n = 48	Severe Disease ^b^ n = 63	*p*
**Haemoglobin initial**	**115–160 g/dL**	**111**	**134 (111–147)**	**139 (129–150)**	**131 (120–145)**	**0.02**
**Haemoglobin lowest**	**115–160 g/dL**	**111**	**112 (98–123)**	**122 (110–129)**	**103 (86–119)**	**<0.0001**
**White cell initial**	**4.0–11.0 × 10^9^/L**	**111**	**9.3 (6.8–11.9)**	**8.2 (6.3–11.0)**	**9.8 (7.2–12.8)**	**0.046**
**White cell highest**	**4.0–11.0 × 10^9^/L**	**111**	**12.8 (9.5–18.3)**	**10.6 (8.1–13.3)**	**15.9 (11.9–22.1)**	**<0.0001**
White cell lowest	4.0–11.0 × 10^9^/L	111	5.7 (4.2–7.8)	5.5 (4.2–6.9)	6.1 (4.2–8.4)	0.32
**Platelet initial**	**140–400 × 10^9^/L**	**111**	**119 (72–165)**	**146 (113–197)**	**84 (47–141)**	**<0.0001**
**Platelet count lowest**	**140–400 × 10^9^/L**	**111**	**84 (33–121)**	**115 (90–147)**	**54 (24–90)**	**<0.0001**
**Neutrophil initial**	**2.0–8.0 × 10^9^/L**	**111**	**8.2 (5.5–10.6)**	**6.7 (4.6–9.3)**	**8.5 (6.3–11.7)**	**0.02**
**Neutrophil highest**	**2.0–8.0 × 10^9^/L**	**111**	**10.7 (8.1–14.2)**	**9.0 (6.1–11.8)**	**12.8 (10.0–19.9)**	**<0.0001**
Neutrophil lowest	2.0–8.0 × 10^9^/L	111	3.5 (2.7–5.6)	3.4 (2.5–4.6)	4.0 (2.8–6.1)	0.10
**Lymphocyte initial**	**1.0–4.0 × 10^9^/L**	**111**	**0.5 (0.3–0.7)**	**0.6 (0.4–0.8)**	**0.4 (0.3–0.6)**	**0.005**
**Lymphocyte lowest**	**1.0–4.0 × 10^9^/L**	**111**	**0.3 (0.2–0.5)**	**0.4 (0.3–0.6)**	**0.3 (0.2–0.4)**	**0.0001**
**INR initial**	**0.9–1.2**	**85**	**1.1 (1.1–1.3)**	**1.1 (1.1–1.1)**	**1.1 (1.1–1.3)**	**0.02**
**INR highest**	**0.9–1.2**	**85**	**1.2 (1.1–1.3)**	**1.1 (1.1–1.1)**	**1.2 (1.1–1.4)**	**0.0002**
APTT initial	25–38 s	111	31 (29–34)	31 (29–33)	31 (28–34)	0.52
**APTT highest**	**25–38 s**	**111**	**32 (30–36)**	**31 (29–33)**	**33 (30–39)**	**0.004**

The median (interquartile range) is presented. Bold text signifies a statistically significant association. INR: international normalised ratio; APTT: activated partial thromboplastin time. ^a^ Queensland public hospital laboratory reference ranges. ^b^ Severe disease is defined as ICU admission or pulmonary haemorrhage.

**Table 4 pathogens-14-00643-t004:** Biochemistry values in individuals admitted to a referral hospital with leptospirosis in Far North Queensland, January 2015–June 2024, and the association of these values with the development of severe disease.

Variable	Reference Range ^a^	Number with Data	All n = 111	No Severe Disease ^b^ n = 48	Severe Disease ^b^ n = 63	*p*
**Initial serum sodium**	**135–145 mmol/L**	**111**	**133 (129–135)**	**134 (130–137)**	**132 (128–135)**	**0.01**
**Lowest serum sodium**	**135–145 mmol/L**	**111**	**132 (129–135)**	**133 (130–136)**	**131 (126–134)**	**0.001**
**Initial serum potassium**	**3.5–5.2 mmol/L**	111	3.7 (3.4–4.0)	3.7 (3.4–4.0)	3.6 (3.4–4.0)	0.55
**Lowest serum potassium**	**3.5–5.2 mmol/L**	**111**	**4.4 (4.0–4.9)**	**4.2 (3.8–4.7)**	**4.6 (4.2–4.9)**	**0.0003**
eGFR initial	>90 mL/min/1.73 m^2^	99	64 (25–90)	77 (27–90)	54 (25–78)	0.12
**eGFR lowest**	**>90 mL/min/1.73 m^2^**	**99**	**38 (14–78)**	**76 (18–90)**	**25 (14–54)**	**0.002**
**Initial serum creatinine**	**45–90 µmol/L**	**111**	**113 (88–205)**	**100 (81–197)**	**124 (93–232)**	**0.04**
**Highest serum creatinine**	**45–90 µmol/L**	**111**	**179 (102–382)**	**107 (85–337)**	**211 (142–433)**	**0.0008**
Initial serum bicarbonate	**22–32 mmol/L**	111	23 (21–25)	24 (22–26)	23 (21–25)	0.07
**Lowest serum bicarbonate**	**22–32 mmol/L**	**111**	**20 (17–22)**	**22 (19–23)**	**19 (16–21)**	**0.0001**
Initial serum bilirubin	**<20 µmol/L**	111	18 (12–28)	16 (10–26)	19 (13–29)	0.19
**Highest serum bilirubin**	**<20 µmol/L**	**111**	**26 (19–48)**	**20 (12–45)**	**29 (21–50)**	**0.004**
Initial serum ALT	<34 IU/mL	111	68 (27–115)	68 (26–126)	67 (27–110)	0.91
Highest serum ALT	<34 IU/mL	111	121 (68–208)	131 (69–189)	120 (68–220)	0.66
Initial serum AST	<31 IU/mL	111	63 (34–135)	67 (32–113)	59 (34–143)	0.37
**Highest serum AST**	**<31 IU/mL**	**111**	**131 (74–210)**	**102 (67–166)**	**153 (80–287)**	**0.02**
Initial serum GGT	<38 IU/mL	111	51 (22–120)	52 (23–159)	48 (22–103)	0.66
Highest serum GGT	<38 IU/mL	111	135 (70–235)	142 (69–297)	135 (70–217)	0.57
Initial serum SAP	30–110 IU/mL	111	98 (67–171)	113 (70–164)	89 (64–174)	0.49
Highest serum SAP	30–110 IU/mL	111	146 (109–208)	143 (114–221)	152 (91–208)	0.88
Initial serum LDH	120–250 IU/mL	111	296 (236–359)	295 (238–358)	296 (234–386)	0.66
**Highest serum LDH**	**120–250 IU/mL**	**111**	**400 (327–534)**	**359 (280–401)**	**480 (377–690)**	**<0.0001**
**Initial serum CK**	**34–145 IU/mL**	**83**	**281 (104–1020)**	**124 (73–438)**	**486 (118–1160)**	**0.01**
**Highest serum CK**	**34–145 IU/mL**	**83**	**350 (114–1020)**	**124 (73–438)**	**621 (124–1225)**	**0.004**
**Initial serum CRP**	**<5 mg/L**	**107**	**190 (138–287)**	**156 (102–234)**	**233 (165–323)**	**0.002**
**Highest serum CRP**	**<5 mg/L**	**107**	**227 (159–323)**	**187 (137–238)**	**258 (195–353)**	**0.0002**
**Initial serum lactate**	**0.5–2.2 mmol/L**	**105**	**1.5 (1.1–2.3)**	**1.3 (1.0–1.8)**	**1.7 (1.2–2.4)**	**0.02**
**Highest serum lactate**	**0.5–2.2 mmol/L**	**105**	**2.0 (1.4–2.7)**	**1.5 (1.2–2.3)**	**2.2 (1.8–3.6)**	**0.0001**
**Elevated initial serum troponin ^c^**	**-**	**111**	**25/111 (23%)**	**2/48 (4%)**	**23/63 (37%)**	**<0.0001**
**Elevated serum troponin during hospitalisation ^c^**	**-**	**111**	**29/111 (26%)**	**2/48 (4%)**	**27/63 (43%)**	**<0.0001**

The median (interquartile range) or the absolute number (percentage) is presented. Bold text signifies a statistically significant association. ALT: alanine aminotransferase; AST: aspartate aminotransferase; GGT: gamma glutamyl transferase; SAP: serum alkaline phosphatase; LDH: lactate dehydrogenase; CK: creatinine kinase; CRP: C-reactive protein. ^a^ Queensland public hospital laboratory reference ranges. ^b^ Severe disease is defined as ICU admission or pulmonary haemorrhage. ^c^ Troponin is expressed as a categorical variable as the Beckman Coulter assay for troponin I (normal reference range <0.040 µg/L) was replaced by the Siemens Atellica assay (reference range <20 ng/L normal for men; <10 ng/L normal for women) during the study period.

**Table 5 pathogens-14-00643-t005:** Chest imaging findings in individuals admitted to a referral hospital with leptospirosis in Far North Queensland, January 2015–June 2024, and the association of these findings with the development of severe disease.

Variable	Number with Data	All n = 111	No Severe Disease ^a^ n = 48	Severe Disease ^a^ n = 63	*p*
**Abnormal initial chest imaging**	**109**	**37/109 (34)**	**10/46 (22)**	**27 (43)**	**0.02**
**Any abnormal chest imaging during hospitalisation**	**109**	**63/109 (58)**	**17/46 (37)**	**46 (73)**	**<0.001**
**Multilobar involvement**	**109**	**49/109 (45)**	**12/46 (26)**	**38 (60)**	**<0.001**
**Alveolar changes**	**109**	**54/109 (49)**	**10/46 (22)**	**44 (70)**	**<0.001**
**Interstitial changes**	**109**	**25/109 (23)**	**6/46 (13)**	**19 (30)**	**0.04**
Pleural effusion	109	19/109 (17)	8/46 (17)	12 (19)	1.00

Absolute numbers (percentages) are presented. Bold text signifies a statistically significant association. ^a^ Severe disease is defined as ICU admission or pulmonary haemorrhage.

## Data Availability

Data cannot be shared publicly because of the Queensland Public Health Act 2005. Data are available from the Far North Queensland Human Research Ethics Committee (contact via email FNQ_HREC@health.qld.gov.au) for researchers who meet the criteria for access to confidential data.

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
