# Peer review of "Presentation and Clinical Course of Leptospirosis in a Referral Hospital in Far North Queensland, Tropical Australia"

_pathogens, 2025, doi:10.3390/pathogens14070643_

Round 1
Reviewer 1 Report
Comments and Suggestions for Authors
The article describes hospitalized leptospirosis cases in Queensland, Australia. It demonstrates a low mortality with adequate care. This is an important finding. However, there are many methodology issues and rules of scientific writing are ignored. The article can be shortened. The senior authors could have corrected language and style before submission. I only recommend publication after major revision.
Review of “Presentation and Clinical Course of Leptospirosis in a Referral Hospital in Far North Queensland, Tropical Australia”
Abstract:
Line 16: “Only 13/111 (12%) had any comorbidities”. Would you expect more? If not, I would not use “Only”
Line 17-19: consider not repeating “required” so many times
Intro
38: “Mammals are the primary reservoir for Leptospira organisms which can survive for many months in urine-contaminated soil and fresh water.”
A bit misleading. Sounds as if the soil requires urine for the Leptospires to survive. Consider revising
42: “Infection commonly occurs after occupational or recreational exposure”.
Not only. Persons who live in slums or highly contaminated environments will get exposed in any daily activity. Also mention flooding, a very important exposure.
48: “The case fatality rate of severe leptospirosis can exceed 50% but reported rates vary enormously and are likely to be explained, predominantly, by geographical variation in access to advanced supportive care [1,12-16]. “
Sadly, as well by the capacity of diagnosing the disease in time (very low awareness, no diagnostic tests available).
68 “We hypothesised that the recent local expansion of critical care capacity would have translated into better patient outcomes, justifying the health service’s investment in these services.
A hypothesis needs testing. I could not find an analysis which tested this hypothesis and I could not find the results, which accepted or rejected this hypothesis. Either you need to remove this sentence or try to test your hypothesis.
Methods
85 and further: leptospira is italic
Criteria:
Culture: why only from blood? No urine cultured?
MAT: Why only fourfold rise in titre and not seroconversion (i.e. from negative to 1:100 or 1:200?)
PCR: which PCR? Why not from urine?
Can you go back and check if you missed some patients due to your inclusion criteria?
Please add: As the retrospective data were deidentified and presented in an aggregated manner, the requirement for informed consent had been waived.
Figure 1 does not provide much information. Could be smaller. Maybe show that it is in a tropical zone? Is there a lot of rainfall?
95: twice “were said to 95 have”. Repetition is not so elegant
107 Oliguria was recorded as present if it was documented in the medical record.
All diagnostic testing results for leptospirosis – including PCR, serology, and culture – was sought and the results were recorded.
I assume the tests were done when patients were admitted? It would be interesting to know, which laboratories conducted the lepto testing and what MAT panels and PCRs were used.
116: “frank haemoptysis or if frank blood”: is “frank” used in medical terms? (I am not a native English speaker)?
117 Individuals were said to have classified with severe leptospirosis if they were admitted to the ICU, or they had pulmonary haemorrhage.
Was the decision to admit to ICU left to the individual physician or do they follow SOPs for that decision? This should be described. I wonder whether “severe disease” is the correct expression. Isn`t it rather “life threatening disease”? Even before being admitted to ICU a disease can be “severe”? It depends on the country’s policies as when patients will be referred to ICU...
122: “Data were de-identified, entered into an electronic database (Microsoft Excel and analysed with statistical software (Stata version 18”.
Usually one needs to add the trademark sign.
123 “Groups were analysed using the Wilcoxon rank-sum test, the chi-squared test, Fisher’s exact test or logistic regression, where appropriate”.
Where appropriate is too general. I would like to know which statistical tests were applied for the specific research questions.
The distinction and how remote and rural residence was categorized (as described in results) should be described in the methods section.
Results
135: "There were 111 individuals who satisfied inclusion criteria for the study, 94/111 (85%) were male and 86/111 (77%) were transferred to Cairns Hospital from another health facility".
Were they from one or both catchment areas? Did cases as well come from other areas than the two you mentioned?
138 : “In 81/111 (73%) there was a potential occupational or environmental exposure preceding the presentation”.
I would be interested in more details on occupational and environmental exposure (these are important findings, especially to make recommendations how to protect citizens from getting infected).
139: The number of cases admitted to Cairns Hospital increased between 2015 and 2023, the first and last completed calendar years in the study period (rs=0.75, p=0.02) (Figure 2).
I do not understand the second part of the sentence. Please rephrase. Please also mention in the methods section which statistical test you used and how you exactly compared the different years to each other. Thanks.
Figure 2: legend text uses two different font sizes.
I think you could show in this figure a bit more than just absolute numbers. For example, you could calculate the median of all cases over the whole study period and show this median as a line in the figure. Then I would add the number of ICU admissions per year.
147: The median (interquartile range (IQR)) duration of hospitalisation was 5 (3-8) days. Why not write: The median duration of hospitalisation was 5 and the interquartile range (IQR) 3-8 days.
153: "The median (IQR) age of the cohort was 38 (24-55) years; there were only 6 children in the cohort and only one of these children was aged less than 10 years."
I would start the results section with the description of the population, including gender (as you did) and age. Further, “only” is an interpretation, however, in the results section we usually just describe. So, I would leave out the “only” in the results. However, in the discussion you can mention that there were more adults than children in the study population.
154 “Severe disease was more common in older patients, but 40/63 (63%) of patients with severe disease were under the age of 50”.
First you need to define the age groups and show them in table 1. “Older” is not precise. In some low-income countries 50 is old, in Japan 80 is old. Report by age-group, then you can show the differences, if they are statistically significantly different. Every difference you show, needs to be tested statistically.
Table 1:
The definition of severe leptospirosis should be described under table 1 (if you keep this definition).
Show age groups and sex with female and male categories. You need to illustrate in the table that you present age by the median and IQR.
Dear senior author of this paper, please make sure you spot these little errors before the first author submits the article. Reviewers should be able to concentrate on the content, not form.
The classification of remote and rural residence should be described in the methods section.
Change “Wet season presentation” to number of cases in wet season”
Please write the full names of the serovars
156: “There were 13/111 (12%) who had a documented comorbidity, but there was no association between the presence of comorbidity and severe disease, and 53/63 (84%) with severe disease had no comorbidity”.
Please revise this sentence, I had to read it 5 times before it made sense.
166: I would start with your inclusion of cases based on your case definition. How many cases were included based on culture, how many on PCR, how many on MAT (four fold etc.). Then it would be interesting to know how many cases met several of your criteria (i.e. PCR and culture).
168 Convalescent serology was collected in 51/111 (47%).
Only collected or as well tested?
“This was frequently to define the infecting serovar, but in 12/51 (24%) it established the diagnosis of leptospirosis. It was possible to determine the serovar in 59 individuals; the most common were Zanoni (21/59, 36%) and 171 Australis (12/59, 21%)”. The first time in the text, you need to write the full name of the serovar. Also, it is better to write the serogroup and then the serovar.
172 “A greater proportion of individuals with confirmed infection with the Zanoni serovar had severe disease than individuals infected with a confirmed infection with other serovars, but in this modestly sized cohort the difference failed to reach statistical significance (table 1)”. Please show p-value
179 The most common symptoms on presentation were subjective fevers, myalgia, headache,and nausea and vomiting. Twice “and” is not pretty
180 “It was notable that only 10/26 (38%) individuals presenting directly to Cairns Hospital were febrile (temperature ≥ 38°C) and that a temperature ≥ 38°C had been documented previously in only 51/85 (60%) individuals who were transferred to Cairns Hospital from another facility.”
While I totally agree with you, in results we usually present the results objectively and in the discussion you can make your point and interpret this result. Please revise the sentence accordingly.
185 “Patients with hypotension, tachycardia, oliguria, tachypnoea and a requirement for oxygen supplementation on their initial assessment at Cairns Hospital were also more likely to develop severe disease”.
Please add p-values
Table 2: a legend should include a short description of the population under investigation and what you describe. “Number and proportion of patients with.....”. Please write n(%) at the top in the table. Then you can leave away the % sign in all rows. It should not be me telling you this...
200 “Patients presenting with acute kidney injury were more likely to develop severe disease than patients without acute kidney injury (33/49 (67%) versus 30/62 (48%), OR (95% CI): 2.20 (1.01-4.79), p=0.047).”
Nice finding but please revise sentence (English). Same for the sentences following until next table.
Table 3 and 4: see comments for Table 2. Please add a column showing the range of “normal” values. Are you showing the median and the IQR? Please, you need to explain in the table, what you are illustrating.
How come you show these massive tables without mentioning hardly any results in the text? The CRP finding seems quite interesting? And I am sure there is more, but I am not a clinician.
225: Individuals with abnormal chest imaging at presentation were more likely to develop severe disease during their hospitalisation than those with normal 226 chest imaging (27/37 (73%) versus 36/72 (50%), OR (95% CI): 2.70 (1.14-6.38), p=0.02).
Please revise sentence. Whenever you describe an association please write the p-value in the text. It is cumbersome to go to the table and check the p-value.
256n“Of the patients admitted to ICU, 38/56 (68%) required vasopressor support, 14/56 256 (25%) required intubation and mechanical ventilation and 13/56 (23%) required RRT”
Replace at least one required with another word.
From line 257 – 266 you use the word “required” 7 times. I know this is not a poetry competition, but please make a bit more effort in English writing.
Figure 3. I am not a clinician, nevertheless I wonder whether the info in Figure 3 is important enough for the space it is taking. Please explain why you think this is so important. The font size is not the same
341: “As in prior studies leptospirosis was diagnosed more commonly in young men: 85% 340 of the cohort was male and almost 70% were younger than 50 years of age, which is no doubt explained by the greater likelihood of occupational and recreational exposure to the pathogen in these populations [1,2].. “
Using the expression “No doubt” is dangerous in a scientific article, especially if you did not mention the exposures in your results. Most studies show leptospirosis in older men, not younger.
397 “The study again highlights the value of PCR in the prompt diagnosis of leptospirosis [55]. PCR had a sensitivity of almost 90% in our cohort compared with a sensitivity of 43% for serology when these tests were collected early in the patient’s hospitalisation.”
I cannot recall a calculation of sensitivity and specificity of the different methods in this article. There was not a description of it in methods nor a presentation of it in results. The word sensitivity in epidemiology is used to calculate the performance of a diagnostic test. I did not see this calculation in your article. Please remove the word “sensitivity” in the above text.
Conclusions: in settings with low resources, early recognition and prompt treatment is key. Probably your findings cannot be translated and need a different approach. Consider revising your last sentence
Comments on the Quality of English LanguagePlease improve the scientific writing style and English
Author Response
Reviewer 1
Comments and Suggestions for Authors
The article describes hospitalized leptospirosis cases in Queensland, Australia. It demonstrates a low mortality with adequate care. This is an important finding. However, there are many methodology issues and rules of scientific writing are ignored. The article can be shortened. The senior authors could have corrected language and style before submission. I only recommend publication after major revision.
Response: We thank Reviewer 1 for the time that he/she has taken to review our manuscript and the very helpful suggestions that he/she has made for its enhancement.
Many of his/her comments were related to style and language, which will, of course, always be subjective. We note that Reviewer 2 described the manuscript as “clear and rather well written”. Reviewer 1 acknowledges that he/she is “not a native English speaker” which may explain some of the concerns. He/she also has some queries about the presentation of the clinical data, while again acknowledging that “I am not a clinician”.
However, clearly many of the readers will (we hope!) be non-native English speakers who may also not be clinicians, so we need to ensure that our manuscript is comprehensible for them.
Reviewer 1’s comments therefore offer us a valuable opportunity to review and revise our paper. We have shortened the paper while also adding information to address his/her concerns. Please find our point-by-point response to his/her comments below.
Review of “Presentation and Clinical Course of Leptospirosis in a Referral Hospital in Far North Queensland, Tropical Australia”
Abstract:
Line 16: “Only 13/111 (12%) had any comorbidities”. Would you expect more? If not, I would not use “Only”
Response: We thank Reviewer 1 for highlighting this issue. Yes, honestly, we might expect that more than 12% of individuals hospitalised at this referral centre would have comorbidities, as individuals with comorbidity would have less physiological reserve and therefore more likely to require escalation of care to a referral centre (rather than being managed as an outpatient or in their local hospital).
The fact that almost 90% of the cohort had no comorbidity (including 47/56 (84%) requiring ICU care) is, we believe, a striking finding as this demonstrates that leptospirosis can cause severe, life-threatening disease in even otherwise well individuals. We have not made any changes to the abstract as this is a point that we think should be highlighted.
Changes: None
Line 17-19: consider not repeating “required” so many times
Response: We thank Reviewer 1 for raising this concern, although this is really an issue of style. As native English speakers we do not feel that repeating the word required is necessarily problematic. However, as it appears to be distracting for the Reviewer and, potentially for other readers, we have substituted “needed” and “received” for “required” on different occasions in the abstract.
Changes: Revision of abstract to prevent frequent repetition of the word “required” (lines 34-37).
Intro
38: “Mammals are the primary reservoir for Leptospira organisms which can survive for many months in urine-contaminated soil and fresh water.”
A bit misleading. Sounds as if the soil requires urine for the Leptospires to survive. Consider revising
Response: We thank Reviewer 1 for raising this issue, but don’t feel that this phrasing suggests that urine is required for the Leptospires to survive. Rather this is how the organism enters the soil and fresh water. We note that the CDC uses very similar phrasing:
“The bacteria can survive for weeks to months in urine-contaminated water and soil.”
However, as it appears to be distracting, we have revised the text.
Changes: Revision of text to prevent confusion about role of urine in the survival of Leptospires (lines 58-59).
42: “Infection commonly occurs after occupational or recreational exposure”.
Not only. Persons who live in slums or highly contaminated environments will get exposed in any daily activity. Also mention flooding, a very important exposure.
Response: We thank Reviewer 1 for raising this issue. In fairness, we don’t say that infection only occurs after occupational or recreational exposure, rather that it commonly occurs after this exposure. Our data (which we have now expanded) show this: in 74/81 (91%) of the cases in the cohort in which a putative exposure could be identified, the exposure was occupational or recreational. There was only 1 case (in this tropical region with monsoonal wet seasons) where flooding was felt to have contributed. However, we are happy to highlight that flooding (and living in rodent-infested, overcrowded urban areas) can also increase the risk of exposure. We have added this in the revised manuscript with an additional citation.
Changes: Revision of text to highlight that leptospirosis also is more common flood-prone, overcrowded urban areas with large rodent populations (lines 62-64 and reference 9).
48: “The case fatality rate of severe leptospirosis can exceed 50% but reported rates vary enormously and are likely to be explained, predominantly, by geographical variation in access to advanced supportive care [1,12-16]. “
Sadly, as well by the capacity of diagnosing the disease in time (very low awareness, no diagnostic tests available).
Response: We thank Reviewer 1 for raising this point. We agree that laboratory capacity is important to establish the diagnosis of leptospirosis and to exclude alternative diagnoses. We have amended the text accordingly (lines 70-73).
However, it is important to note that in most of the patients in this cohort, the diagnosis of leptospirosis was not confirmed until several days the patient’s presentation. The most common way that individuals were diagnosed in this cohort was with PCR which was performed in a reference laboratory in Brisbane 1390 km away (as we now highlight in the methods) and the result was not available for up to a week. Indeed, frequently, the diagnosis of leptospirosis was a retrospective one, that was established with convalescent serology several weeks after the patient’s presentation.
Most first line antibiotics (including penicillins, cephalosporins, macrolides, tetracyclines and fluoroquinolones) used empirically for patients with infection/sepsis will cover the organism. They can receive these antibiotics without a definitive diagnosis. The key for the patient is to be able to access medical care and to be able to receive appropriate medical care.
Changes: Revision of text to highlight that variation in case-fatality rate can also be explained by geographical differences in resourcing of health systems, including local diagnostic capacity (lines 70-73).
68 “We hypothesised that the recent local expansion of critical care capacity would have translated into better patient outcomes, justifying the health service’s investment in these services.
A hypothesis needs testing. I could not find an analysis which tested this hypothesis and I could not find the results, which accepted or rejected this hypothesis. Either you need to remove this sentence or try to test your hypothesis.
Response: We thank Reviewer 1 for raising this issue, we have removed all reference to a hypothesis in the revised manuscript.
Changes: Removal of this sentence, as suggested.
Methods
85 and further: leptospira is italic
Response: We thank Reviewer 1 for highlighting this error. We have amended the text accordingly.
Changes: Use of italics for Leptospira throughout the manuscript.
Criteria:
Culture: why only from blood? No urine cultured?
Response: We thank Reviewer 1 for raising this point. It is not clinical practice to culture urine for Leptospira in our region of Australia. See the Queensland state guidelines for the diagnosis of leptospirosis here:
https://www.health.qld.gov.au/cdcg/index/lepto#:~:text=For%20suspected%20leptospirosis%20cases%20presenting,3.
The organism is cultured in the state reference laboratory in Brisbane which is over 1000km away from Cairns hospital and so the yield of culture from urine is low, particularly as the acid pH of urine decreases the viability of the organism. There is also a high contamination rate.
https://www.health.gov.au/sites/default/files/2025-01/leptospirosis-laboratory-case-definition.pdf
I have liaised with Dr Megan Staples the Senior Scientist from the Leptospirosis Reference Laboratory and WHO Collaborating Centre for Reference and Research on Leptospirosis (Western Pacific Region) to confirm this; her advice is now acknowledged in the paper.
Another reason that culture of urine in our region would have a low yield is that in the natural history of the disease, the organism is usually present in urine in higher numbers in the second week of disease (https://pmc.ncbi.nlm.nih.gov/articles/PMC4442676/). Our patients will have almost always have received antibiotics by then, further reducing the yield of culture.
Changes: We have added text to highlight that urinary culture for Leptospira organisms is not performed in Queensland.
MAT: Why only fourfold rise in titre and not seroconversion (i.e. from negative to 1:100 or 1:200?)
Response: We thank Reviewer 1 for raising this point. We used the Australian laboratory criteria for a laboratory confirmed diagnosis of leptospirosis to define our study population.
https://www.health.gov.au/sites/default/files/2025-01/leptospirosis-laboratory-case-definition.pdf
The lowest titre for our MAT is < 1:50, so titres in paired sera of <1: 50 and 1:200 would represent a fourfold rise and would be captured in this definition. titres in paired sera of <1: 50 and 1:100 would not represent a fourfold rise and would therefore be captured in this definition. I have liaised with Dr Megan Staples the Senior Scientist from the Leptospirosis Reference Laboratory and WHO Collaborating Centre for Reference and Research on Leptospirosis (Western Pacific Region) to confirm this.
We have used the strictest definition of laboratory-confirmed disease to ensure that we are only describing individuals with definite leptospirosis (in a region where leptospirosis is an endemic pathogen).
Changes: None
PCR: which PCR? Why not from urine?
Response: We thank Reviewer 1 for raising this issue. As outlined above testing of urine for leptospirosis (with either culture or PCR) is not part of the state guidelines for diagnosis of leptospirosis in the state of Queensland.
https://www.health.qld.gov.au/cdcg/index/lepto#:~:text=For%20suspected%20leptospirosis%20cases%20presenting,3.
For most of the study period the Leptospirosis reference laboratory did not have National Association of Testing Authorities (NATA) accreditation for PCR of urine; this has only recently been granted (13 May 2025). I have liaised with Dr Megan Staples the Senior Scientist from the Leptospirosis Reference Laboratory and WHO Collaborating Centre for Reference and Research on Leptospirosis (Western Pacific Region) to confirm this.
Changes: We have added text to highlight that urinary PCR for Leptospira organisms was not available during the study period.
Can you go back and check if you missed some patients due to your inclusion criteria?
Response: We thank Reviewer 1 for raining this issue. We performed an automated search of our electronic laboratory database (AUSLAB) as described in the methods (lines 107-108). All positive results were cross checked manually by review of the patient charts and the department of infectious diseases database. As urine culture and urine PCR for Leptospira were not performed during the study period in our region, we do not believe that there are confirmed cases that have been missed.
Changes: None
Please add: As the retrospective data were deidentified and presented in an aggregated manner, the requirement for informed consent had been waived.
Response: We thank Reviewer 1 for raising this issue. In the initial submission, we did write “As the retrospective data were deidentified and presented in an aggregated manner, the Committee waived the requirement for informed consent.” Is this OK?
In a prior publication in Pathogens (pathogens-3588579) earlier this year we were asked by the Editorial staff to present our Institutional Review Board Statement and Informed Consent Statement at the end of the document. We are happy for this information to be presented in the article wherever the Editor feels it is best located.
Changes: None
Figure 1 does not provide much information. Could be smaller. Maybe show that it is in a tropical zone? Is there a lot of rainfall?
Response: We thank Reviewer 1 for their constructive feedback. Australia is a big country with a variety of climates. Far North Queensland (the study area) is a region of 380,000km2 over five times the size of Sri Lanka. We feel that the map is helpful for conveying this information. We feel that it is also helpful to distinguish the two health services (one of which we use to define remote residence).
We have already stated that the region is in a tropical location, so we don’t feel that adding the Tropic of Capricorn to the map is helpful, however in the revised manuscript we have added additional data in the methods about the region’s geography and climate (including rainfall) (lines 96-99). We are happy to the Editorial staff to reduce the size of the figure as necessary. We would also be happy to relegate it to a supplementary file if the Editor feels that it adds little for the reader.
Changes: Revision of methods section to provide more geographical data and information about the rainfall in the region (lines 96-99).
95: twice “were said to 95 have”. Repetition is not so elegant
Response: This is largely a question of style. As native English speakers we would argue that it is more appropriate to use the same phrase. However, as it appears to be distracting, we have substituted “were defined as having” for “were said to have” on several occasions in the methods.
Changes: Revision of methods to prevent frequent repetition of the phrase “were said to have”.
107 Oliguria was recorded as present if it was documented in the medical record.
All diagnostic testing results for leptospirosis – including PCR, serology, and culture – was sought and the results were recorded.
I assume the tests were done when patients were admitted? It would be interesting to know, which laboratories conducted the lepto testing and what MAT panels and PCRs were used.
Response: We thank Reviewer 1 for raising this issue. Yes, oliguria at presentation was noted. We have revised the text to make this clearer.
As we note in the limitations section the investigations were not collected in a standardised manner. While investigations were usually performed early in the admission (often on the day of presentation), this was not always the case. Indeed, one of the things that we examined in the study was the “worst” laboratory value during the hospitalisation (which often developed subsequently). We also examined evolution of radiological changes.
Testing for leptospirosis was perhaps the laboratory investigation that varied most. As we have highlighted, PCR was performed on blood in 99/111, whole blood culture was performed in 48/111, acute serology testing was performed in 104/111 (94%) and convalescent serology was tested in 51/111 (47%). The timing of testing also varied significantly and depended, to a large extent, on the clinical suspicion of the attending physician.
ELISA testing was performed at the Cairns hospital local laboratory, while PCR, MAT testing and culture were performed at the Leptospirosis Reference Laboratory and WHO Collaborating Centre for Reference and Research on Leptospirosis (Western Pacific Region). We have added text to clarify this. We have added the target for the PCR testing (outer membrane protein LipL32) and we have also added a supplementary table that describes the MAT panel (supplementary table 1)
Changes: Revision of methods to improve clarity on diagnostic strategies. Addition of a supplementary table to describe the MAT panel.
116: “frank haemoptysis or if frank blood”: is “frank” used in medical terms? (I am not a native English speaker)?
Response: We thank Reviewer 1 for querying this point. Yes, “frank haemoptysis” is a medical term. It distinguishes the expectoration of frank blood from blood streaked or blood-stained sputum.
Changes: None
117 Individuals were said to have classified with severe leptospirosis if they were admitted to the ICU, or they had pulmonary haemorrhage.
Was the decision to admit to ICU left to the individual physician or do they follow SOPs for that decision? This should be described. I wonder whether “severe disease” is the correct expression. Isn`t it rather “life threatening disease”? Even before being admitted to ICU a disease can be “severe”? It depends on the country’s policies as when patients will be referred to ICU...
Response: We thank Reviewer 1 for raising this issue. There are, of course, many different ways to describe more serious disease, but severe disease, we feel, is a reasonable one. There is no standard definition for “severe leptospirosis” which is why we define it on the methods (different authors have used different definitions over the years).
Individuals were admitted to the ICU based on clinical need. As the Cairns Hospital is a public hospital, the 16 ICU beds are reserved for the sickest individuals in the hospital, whatever the aetiology of their life-threatening illness (leptospirosis, septic shock, multi-trauma, rupture of an aortic aneurysm etcetera). There is no national “policy” for ICU admission; individuals are admitted to the ICU if beds are available and if they have a requirement for care that cannot be delivered on the ward (including mechanical ventilation and prolonged vasopressor support) or they have monitoring requirements that necessitate higher nurse: patient ratios. These are fairly standard criteria for ICU admission around the world.
Changes: None
122: “Data were de-identified, entered into an electronic database (Microsoft Excel and analysed with statistical software (Stata version 18”.
Usually one needs to add the trademark sign.
Response: We thank Reviewer 1 for raising this issue. In our publication in Pathogens (pathogens-3588579) earlier this year we did not use the trademark sign for either Microsoft Excel or Stata, but we are happy to include this if it is Journal policy.
Changes: None
123 “Groups were analysed using the Wilcoxon rank-sum test, the chi-squared test, Fisher’s exact test or logistic regression, where appropriate”.
Where appropriate is too general. I would like to know which statistical tests were applied for the specific research questions.
Response: We thank Reviewer 1 for highlighting this issue. All four tests are commonly used statistical tests, and we believe that further description of which of these tests we used in which of the multiple comparisons may be unnecessarily inclusive (the reviewer actually suggested shortening the paper in his/her comments).
The Wilcoxon rank test a non-parametric statistical test used to compare the distributions of two independent samples of continuous variables (in this study, for example, the association between age and severe disease). The chi-squared test is used to determine whether two categorical variables are independent in a large sample size (in this study, for example, the association between gender and severe disease). Fisher’s exact test is used to determine whether two categorical variables are independent in a smaller sample size (in this study, for example, the association between diabetes mellitus and severe disease). Logistic regression was used to determine odds ratios and the 95% confidence interval (in this study, for example, the odds ratio of patients in the ICU receiving broad spectrum antibiotic therapy).
We could add this description to the statistics section if the Editor feels that it would enhance the article, however we feel that this may be unnecessary for the Pathogens readership who will be familiar with these commonly used statistical tests.
Changes: None
The distinction and how remote and rural residence was categorized (as described in results) should be described in the methods section.
Response: We did, in fact, describe this in the methods in the original manuscript.
“Individuals living in the TCHHS were said to have a remote residence and these individuals, and the individuals living in the CHHHS, but outside the city of Cairns, were said to have a rural residence.” The definitions are also highlighted in the footnotes to Table 1.
Changes: None
Results
135: "There were 111 individuals who satisfied inclusion criteria for the study, 94/111 (85%) were male and 86/111 (77%) were transferred to Cairns Hospital from another health facility".
Were they from one or both catchment areas? Did cases as well come from other areas than the two you mentioned?
Response: We thank Reviewer 1 for raising this point. No, all 111 individuals in the cohort lived in one of the two catchment areas. If an individual lived in another catchment area, they would have bee referred to a different referral hospital.
Changes: We have added text to make it clearer that all patients were referred from the two catchment areas (lines 161-162).
138 : “In 81/111 (73%) there was a potential occupational or environmental exposure preceding the presentation”.
I would be interested in more details on occupational and environmental exposure (these are important findings, especially to make recommendations how to protect citizens from getting infected).
Response: We thank Reviewer 1 for raising this issue. We have added a supplementary table that describes the potential occupational or environmental exposure in the 81 individuals in the cohort in whom one could be determined. However, there was a wide variety of putative occupational, recreational and environmental exposures making it challenging to make recommendations to prevent infection.
Changes: Addition of a supplementary table (Supplementary table 3) to describe the potential occupational or environmental exposure in the 81 individuals in the cohort in whom one could be determined.
139: The number of cases admitted to Cairns Hospital increased between 2015 and 2023, the first and last completed calendar years in the study period (rs=0.75, p=0.02) (Figure 2).
I do not understand the second part of the sentence. Please rephrase. Please also mention in the methods section which statistical test you used and how you exactly compared the different years to each other. Thanks.
Response: We thank Reviewer 1 for highlighting their concern here. The study period was from January 2015 to June 2024. Therefore, the first and last completed calendar years in the study period were 2015 and 2023. We have revised the sentence to explain this specifically.
We did, in fact, describe the statistical test that we used to determine trend over time in line 151: “Trends over time were analysed using Spearman’s test for correlation.”
Changes: Revision of this sentence to make it clearer (lines 166-167)
Figure 2: legend text uses two different font sizes.
I think you could show in this figure a bit more than just absolute numbers. For example, you could calculate the median of all cases over the whole study period and show this median as a line in the figure. Then I would add the number of ICU admissions per year.
Response: We thank Reviewer 1 for his/her helpful advice about how to improve Figure 2. We have revised the figure with his/her suggestions in mind. We have added the absolute figures above each bar – rather than just the number of ICU cases – to also accommodate the suggestions of Reviewer 2. It is usual for the footnotes of a figure or table to be smaller than the general text. If the manuscript is accepted for publication, we are happy to be advised by Editorial staff if we need to make further adjustments (font style, font size etcetera) to satisfy Journal policy.
Changes: Revision of Figure 2 as per the suggestions of both Reviewers.
147: The median (interquartile range (IQR)) duration of hospitalisation was 5 (3-8) days. Why not write: The median duration of hospitalisation was 5 and the interquartile range (IQR) 3-8 days.
Response: We thank Reviewer 1 for his/her helpful suggestion. We have amended the text as suggested (lines 171-172).
Changes: Revision of text as per the Reviewer’s suggestion.
153: "The median (IQR) age of the cohort was 38 (24-55) years; there were only 6 children in the cohort and only one of these children was aged less than 10 years."
I would start the results section with the description of the population, including gender (as you did) and age. Further, “only” is an interpretation, however, in the results section we usually just describe. So, I would leave out the “only” in the results. However, in the discussion you can mention that there were more adults than children in the study population.
Response: We thank Reviewer 1 for his/her helpful suggestion. We have amended the presentation of the data in the results section to align with his/her suggestions.
With regards to the Reviewer’s suggestions about the use of the word “only”, we feel that this is largely a matter of style. We feel that using the word only in the results highlights the point, but this single word does not represent “interpretation”. However, we would be happy to remove “only” if the Editor feels that its use is problematic.
Changes: Re-ordering of presentation of data in the results as suggested.
154 “Severe disease was more common in older patients, but 40/63 (63%) of patients with severe disease were under the age of 50”.
First you need to define the age groups and show them in table 1. “Older” is not precise. In some low-income countries 50 is old, in Japan 80 is old. Report by age-group, then you can show the differences, if they are statistically significantly different. Every difference you show, needs to be tested statistically.
Response: We thank Reviewer 1 for raising this issue. When we used the word “older” we were referring to the older members of this cohort. We have already highlighted the association between age and severe disease in the tables and discussion. However, we have added a supplementary figure that presents the complete cohort so that the reader is able to examine the data in more granular detail.
Change: Addition of a supplementary table that shows the association between age and severe disease across the entire cohort (Supplementary Figure 2).
Table 1:
The definition of severe leptospirosis should be described under table 1 (if you keep this definition).
Show age groups and sex with female and male categories. You need to illustrate in the table that you present age by the median and IQR.
Dear senior author of this paper, please make sure you spot these little errors before the first author submits the article. Reviewers should be able to concentrate on the content, not form.
Response: We thank Reviewer 1 for his/her helpful suggestions to improve table 1. We agree that we should have highlighted that the age was presented as median (IQR); this was an omission. We have added our definition of severe leptospirosis to the footnotes of this table.
There was no association between gender and the clinical phenotype or clinical course, so we don’t feel that there is any value by presenting the data, stratified by sex (indeed there were only 17 females in the entire cohort). However, we would be happy to if the Editor feels that this is an important issue to highlight. We have added an additional supplementary figure (supplementary figure 2) to present the findings by age in more detail.
Change: Addition of a footnote to table 1, as suggested. Addition of a supplementary figure that shows the association between age and severe disease across the entire cohort.
The classification of remote and rural residence should be described in the methods section.
Response: As we note above, we did do this in the original submission (lines 122-124 in the present submission).
Change: None.
Change “Wet season presentation” to number of cases in wet season”
Response: This is largely a question of style. We believe “wet season presentation” is fine here and is internally consistent as we do not say “number of cases of diabetes mellitus” or “number of cases of hazardous alcohol use” in other cells in the table.
Changes: None
Please write the full names of the serovars YYY
Response: We thank Reviewer 1 for raising this issue. We have added the full names of the Serovars as suggested.
Change: Text amended as suggested (lines 189-191).
156: “There were 13/111 (12%) who had a documented comorbidity, but there was no association between the presence of comorbidity and severe disease, and 53/63 (84%) with severe disease had no comorbidity”.
Please revise this sentence, I had to read it 5 times before it made sense.
Response: We thank Reviewer 1 for highlighting the difficulty with understanding this phrase. We have revised the presentation of these data to facilitate comprehension.
Change: Revision of text (lines 180-182)
166: I would start with your inclusion of cases based on your case definition. How many cases were included based on culture, how many on PCR, how many on MAT (four fold etc.). Then it would be interesting to know how many cases met several of your criteria (i.e. PCR and culture).
Response: We thank Reviewer 1 for raining this issue. As we highlight in the limitations, investigations were not collected in a standardised manner and so the different patients had different tests performed at different times in their illness (which is representative of the real world setting of the study). The patients could be included in the study if they satisfied any of the Australian criteria for laboratory confirmed disease.
However, as the collection of testing was so heterogeneous (blood PCR was performed in 99, while paired serology was performed in only 51 and culture was performed in only 48) and was performed at different stages of their illness (before and after receipt of antibiotics, for instance) we don’t feel that the presentation of these data would be meaningful.
Change: No change.
168 Convalescent serology was collected in 51/111 (47%).
Only collected or as well tested?
Response: We thank Reviewer 1 for highlighting this ambiguity. We have revised “collected” to “collected and tested” as suggested.
Change: Amended text (lines 186-187).
“This was frequently to define the infecting serovar, but in 12/51 (24%) it established the diagnosis of leptospirosis. It was possible to determine the serovar in 59 individuals; the most common were Zanoni (21/59, 36%) and 171 Australis (12/59, 21%)”. The first time in the text, you need to write the full name of the serovar. Also, it is better to write the serogroup and then the serovar.
Response: We thank Reviewer 1 for raising this issue. In the revised manuscript we have written the serogroup and the full name of the serovar.
Change: Amended text as per the Reviewer’s suggestions.
172 “A greater proportion of individuals with confirmed infection with the Zanoni serovar had severe disease than individuals infected with a confirmed infection with other serovars, but in this modestly sized cohort the difference failed to reach statistical significance (table 1)”. Please show p-value
Response: We thank Reviewer 1 for raising this point. We do show the p value for this association in table 1 (as we highlight). Journals usually actively discourage duplication of data in tables and text, so we have not done that in the manuscript. It is important to remember that if the paper is accepted for publication, the table will appear after the paragraph where it is first cited and so it will be easier for the reader to see this association.
Change: No change.
179 The most common symptoms on presentation were subjective fevers, myalgia, headache,and nausea and vomiting. Twice “and” is not pretty
Response: We thank Reviewer 1 for highlighting this point. Nausea and vomiting often coexist and so we presented them together here. But we agree, we can remove the unattractive “and” here and preserve the meaning of the sentence.
Change: Amended text as per the Reviewer’s suggestion.
180 “It was notable that only 10/26 (38%) individuals presenting directly to Cairns Hospital were febrile (temperature ≥ 38°C) and that a temperature ≥ 38°C had been documented previously in only 51/85 (60%) individuals who were transferred to Cairns Hospital from another facility.”
While I totally agree with you, in results we usually present the results objectively and in the discussion you can make your point and interpret this result. Please revise the sentence accordingly.
Response: This is largely a question of style; we are highlighting that, surprisingly, many patients in the cohort were not febrile at presentation. We are happy to rephrase this if the Editor feels that this represents too much interpretation for the results section.
Change: No change.
185 “Patients with hypotension, tachycardia, oliguria, tachypnoea and a requirement for oxygen supplementation on their initial assessment at Cairns Hospital were also more likely to develop severe disease”.
Please add p-values
Response: As we note above, journals usually actively discourage duplication of data in tables and text, so we have not done that in the manuscript. The inclusion of too many numbers in the text of the results can impair legibility. It is important to remember that if the paper is accepted for publication, the table will appear after the paragraph where it is first cited and so it will be easier for the reader to see this association.
Change: No change.
Table 2: a legend should include a short description of the population under investigation and what you describe. “Number and proportion of patients with.....”. Please write n(%) at the top in the table. Then you can leave away the % sign in all rows. It should not be me telling you this...
Response: We are sorry that Reviewer 1 appears to have taken such exception to the presentation of these data. We have presented the data in the table in an identical manner to the way that we presented the data in a prior publication in Pathogens (pathogens-3588579) earlier this year. We note that Reviewer 2 expressed no reservations about the way we presented our data. We would be happy to format the data in any way that the Editor wishes.
Change: No change.
200 “Patients presenting with acute kidney injury were more likely to develop severe disease than patients without acute kidney injury (33/49 (67%) versus 30/62 (48%), OR (95% CI): 2.20 (1.01-4.79), p=0.047).”
Nice finding but please revise sentence (English). Same for the sentences following until next table.
Response: We are uncertain what is concerning the Reviewer about the English of this sentence. However, we have amended it in a manner that may be easier to understand. If Reviewer 1 provides more specific feedback, we would be happy to try and address his/her concerns further.
Change: Amended text (lines 214-217).
Table 3 and 4: see comments for Table 2. Please add a column showing the range of “normal” values. Are you showing the median and the IQR? Please, you need to explain in the table, what you are illustrating.
Response: We thank Reviewer 1 for raising this issue. We have added the reference range to the table. We have also added text saying that we are presenting the median and IQR; we apologise for not presenting this point in the original submission.
Change: Revision of tables 3 and 4 as per the suggestions of the Reviewer.
How come you show these massive tables without mentioning hardly any results in the text? The CRP finding seems quite interesting? And I am sure there is more, but I am not a clinician.
Response: As we note above, journals usually actively discourage duplication of data in tables and text, so we have not done that in the manuscript. The inclusion of too many numbers in the text of the results can impair legibility. It is important to remember that if the paper is accepted for publication, the table will appear after the paragraph where it is first cited and so it will be easier for the reader to see this association.
As expected, there are multiple abnormalities in the haematological and biochemical indices that were tested in individuals hospitalised with leptospirosis at this referral centre; most are to be expected and are common in individuals presenting with leptospirosis, although, as we note, liver function tests were less deranged in our cohort than in other studies (which, as we note in the discussion, may be due to different infecting serovars).
The elevated C-reactive protein is not surprising in individuals with leptospirosis and has been observed previously (https://pmc.ncbi.nlm.nih.gov/articles/PMC6735690/). The fact that it was higher in patients with more life-threatening disease is also not surprising for a disease that is characterised by systemic inflammation. We do plan, as noted in the discussion, to examine in detail, the presentation and management of the patients in his cohort with cardiac involvement, respiratory involvement and a requirement for ICU admission in an effort to define these clinical phenotypes in more detail (lines 409-413)
We also note that the Reviewer suggested shortening the article in his/her comments. Expanded discussion of laboratory findings would tend to lengthen the paper.
Change: No change
225: Individuals with abnormal chest imaging at presentation were more likely to develop severe disease during their hospitalisation than those with normal 226 chest imaging (27/37 (73%) versus 36/72 (50%), OR (95% CI): 2.70 (1.14-6.38), p=0.02).
Please revise sentence. Whenever you describe an association please write the p-value in the text. It is cumbersome to go to the table and check the p-value.
Response: We are uncertain what is concerning the Reviewer about the English of this sentence. However, we have amended it in a manner that may be easier to understand. If Reviewer 1 provides more specific feedback, we would be happy to try and address his/her concerns further.
As we note above, journals usually actively discourage duplication of data in tables and text, so we have not done that in the manuscript. The inclusion of too many numbers in the text of the results can impair legibility. It is important to remember that if the paper is accepted for publication, the table will appear after the paragraph where it is first cited and so it will be easier for the reader to see this association.
Change: Amended text (lines 230-232).
256n“Of the patients admitted to ICU, 38/56 (68%) required vasopressor support, 14/56 256 (25%) required intubation and mechanical ventilation and 13/56 (23%) required RRT”
Replace at least one required with another word.
Response: This is largely a question of style. We (and Reviewer 2) feel this reads OK. However, as it appears to be distracting, we have substituted “needed” and “received” for “required” at different points in the text.
Change: Amended text (lines 262-267).
From line 257 – 266 you use the word “required” 7 times. I know this is not a poetry competition, but please make a bit more effort in English writing.
Response: We appreciate that Reviewer 1 is a little frustrated with our use of language here. This is largely a question of style. We (and Reviewer 2) feel this reads OK. However, as it appears to be distracting, we have substituted “needed” and “received” for “required” at different points in the text.
Change: Amended text (lines 262-267).
Figure 3. I am not a clinician, nevertheless I wonder whether the info in Figure 3 is important enough for the space it is taking. Please explain why you think this is so important. The font size is not the same
Response: We thank Reviewer 1 for raising this issue. Yes, we feel that it is valuable to present this data. The aim of the study was to define the clinical phenotype of individuals with leptospirosis admitted to this referral hospital (as we state in lines 87-88). Frequently severe disease affects multiple organs concurrently and we feel that this figure captures that. We feel that clinicians reading the paper (the primary target readership of the paper) would find these data interesting.
It is usual for the footnotes of a figure or table to be smaller than the general text. If the manuscript is accepted for publication, we are happy to be advised by Editorial staff if we need to make further adjustments (font style, font size etcetera) to satisfy Journal policy.
Change: No change.
341: “As in prior studies leptospirosis was diagnosed more commonly in young men: 85% 340 of the cohort was male and almost 70% were younger than 50 years of age, which is no doubt explained by the greater likelihood of occupational and recreational exposure to the pathogen in these populations [1,2].. “
Using the expression “No doubt” is dangerous in a scientific article, especially if you did not mention the exposures in your results. Most studies show leptospirosis in older men, not younger.
Response: We thank Reviewer 1 for raising this issue. In our region leptospirosis is more common in young men as they are likely to be working or playing in environments where they encounter the pathogen (e.g. fruit picking, framing, water sports etcetera). We have demonstrated this previously in a larger cohort (https://journals.plos.org/plosntds/article?id=10.1371/journal.pntd.0007205) where the median age was 33 (23-45) and 90% were male.
We have, in the revised submission, added more data about exposures that we feel justifies our assertion (supplementary table 3), however, we have softened the language around this, removing the phrase “no doubt”. We have also added a supplementary figure that provides more granular detail about the age of the patients in the cohort.
Change: Addition of a supplementary table that shows the putative exposures and a supplementary figure that shows the distribution of age (and its association with severe disease) across the entire cohort.
397 “The study again highlights the value of PCR in the prompt diagnosis of leptospirosis [55]. PCR had a sensitivity of almost 90% in our cohort compared with a sensitivity of 43% for serology when these tests were collected early in the patient’s hospitalisation.”
I cannot recall a calculation of sensitivity and specificity of the different methods in this article. There was not a description of it in methods nor a presentation of it in results. The word sensitivity in epidemiology is used to calculate the performance of a diagnostic test. I did not see this calculation in your article. Please remove the word “sensitivity” in the above text.
Response: We thank Reviewer 1 for raining this issue. We calculated sensitivity by determining the number of individuals with laboratory confirmed leptospirosis in whom PCR was tested (99) and the proportion in whom it was positive (89); 88/99=89.9%
Change: No change
Conclusions: in settings with low resources, early recognition and prompt treatment is key. Probably your findings cannot be translated and need a different approach. Consider revising your last sentence
Response: We thank Reviewer 1 for their suggestion. We have revised the conclusions on line with the Reviewer’s suggestions.
Change: Revision of conclusions to address the Reviser’s concerns (lines 420-424).
Reviewer 2 Report
Comments and Suggestions for Authors
This manuscript describes a retrospective cohort study related to leptospirosis, a zoonotic disease caused by the pathogenic Leptospira spp. This study included 111patients with diagnosed leptospirosis admitted to Cairns Hospital (a referral hospital) in Australia, between January 2015 and June 2024. The primary focus was on identifying factors that may influence the course of leptospirosis and its severity. Among 111 patients, 63 (57%) presented with severe disease. Importantly, every patient in the cohort survived to hospital discharge. This was attributed to Australia’s well-resourced health care system providing access to multimodal critical care support and also the awareness of leptospirosis in this part of the world.
The manuscript is clear and rather well written, and presents important epidemiological and clinical data.
I only have a few minor comments and suggestions for the authors.
- The authors cite the number of cases of leptospirosis occurring in the world (lanes 33-35). How many cases of leptospirosis are reported in Australia each year? Is there such data?
- Figure 2: I would suggest putting the number of cases of laboratory-confirmed leptospirosis above the bars, because it is not very easy to read this number from the graph. This suggestion also applies to Figure 4.
- Lanes 253-254: The authors mentioned the occurrence of Jarisch-Herxheimer reaction in one of the patients in the cohort. Was this a severe case or did the reaction resolve spontaneously? Does this reaction occur frequently in patients with leptospirosis undergoing antibiotic therapy?
Author Response
Reviewer 2
This manuscript describes a retrospective cohort study related to leptospirosis, a zoonotic disease caused by the pathogenic Leptospira spp. This study included 111patients with diagnosed leptospirosis admitted to Cairns Hospital (a referral hospital) in Australia, between January 2015 and June 2024. The primary focus was on identifying factors that may influence the course of leptospirosis and its severity. Among 111 patients, 63 (57%) presented with severe disease. Importantly, every patient in the cohort survived to hospital discharge. This was attributed to Australia’s well-resourced health care system providing access to multimodal critical care support and also the awareness of leptospirosis in this part of the world.
The manuscript is clear and rather well written, and presents important epidemiological and clinical data.
Response: We thank Reviewer 2 for the time that he/she has taken to review our manuscript, the encouraging feedback and the helpful suggestions that he/she has made for its enhancement. Please find our point-by-point response to his/her comments below.
I only have a few minor comments and suggestions for the authors.
- The authors cite the number of cases of leptospirosis occurring in the world (lanes 33-35). How many cases of leptospirosis are reported in Australia each year? Is there such data?
Response: We thank Reviewer 2 for raising this issue. Yes, leptospirosis is a nationally notifiable disease. Cases are monitored through the National Notifiable Diseases Surveillance System. https://nindss.health.gov.au/pbi-dashboard/
There were, for instance, 140 cases of laboratory confirmed human leptospirosis in 2024 (although this figure is likely to underestimate the true incidence as many cases are treated empirically without complete testing)
Change: We have added a reference to Australia’s National Notifiable Diseases Surveillance System (and its publicly available data) for the interested reader (reference 19).
- Figure 2: I would suggest putting the number of cases of laboratory-confirmed leptospirosis above the bars, because it is not very easy to read this number from the graph. This suggestion also applies to Figure 4.
Response: We thank Reviewer 2 for this suggestion. We have revised Figure 2 and Figure 4 as suggested (while also incorporating suggestions from Reviewer 1).
Change: Revision of Figure 2 as per the suggestions of both Reviewers.
- Lanes 253-254: The authors mentioned the occurrence of Jarisch-Herxheimer reaction in one of the patients in the cohort. Was this a severe case or did the reaction resolve spontaneously? Does this reaction occur frequently in patients with leptospirosis undergoing antibiotic therapy?
Response: We thank Reviewer 2 for highlighting this issue. This 45-year-old individual presented to a peripheral hospital with a 4-day history of fever, polyarthralgia, lethargy, nausea and vomiting. He was commenced on benzylpenicillin and doxycycline. He deteriorated very shortly after the administration of antibiotics, developing hypotension (systolic blood pressure of 50mmHg) which required fluid resuscitation and the commencement of noradrenaline and hydrocortisone.
He was transferred to Cairns Hospital and admitted to ICU with multi-organ dysfunction (acute kidney injury, myocardial injury and transaminitis). He spent 3 days in ICU (where the only support that he received was vasopressor support; it was able to be weaned rapidly.
Although this case was the only definite Jarisch-Herxheimer reaction that we recorded in this cohort, anecdotally they are not uncommon in the peripheral hospitals where the majority of cases in our region are seen and where the first dose of antibiotics is more usually administered. We do highlight the local incidence of Jarisch-Herxheimer reactions in the discussion and the fact that referral hospital setting of the study is likely to have underestimated its incidence (lines 305-306 and 382-383).
Reviewer 3 Report
Comments and Suggestions for Authors
Stratton et al conducted a study on the presentation and clinical course of leptospirosis in a referral hospital in Far North Queensland, tropical Australia, which is of some significance. The fowlloing aspects can be considered to improve the quarlity of the study.
1. a significant shortcoming in the study is the absence of adequate statistic analyses in all Tables. The authors should add 95% confidence intervals (95% CI) for all occurrence rates presented, and Include more robust statistical analyses for the identified risk factors (age, gender, etc.)
2. figure 4, the number of shared areas need to be inluded.
3. there are quite a few format and spelling issues, such as there should be a blank between number and unit.
Author Response
Stratton et al conducted a study on the presentation and clinical course of leptospirosis in a referral hospital in Far North Queensland, tropical Australia, which is of some significance. The fowlloing aspects can be considered to improve the quarlity of the study.
- a significant shortcoming in the study is the absence of adequate statistic analyses in all Tables. The authors should add 95% confidence intervals (95% CI) for all occurrence rates presented, and Include more robust statistical analyses for the identified risk factors (age, gender, etc.)
Response: We thank the reviewer for the time that he/she has taken to review our manuscript. We are happy to hear that he/she feels that paper is of some significance.
As this is a case series which simply reports the proportion of individuals with a given finding (gender, comorbidity, symptoms, signs, imaging abnormalities) we don’t feel that adding 95% confidence intervals has any value for the reader. Indeed, we feel that this would make the tables more cluttered and less readable. Indeed, another reviewer has asked to remove the percentage symbols from the table for this very reason (which we have). Adding confidence intervals for all proportions would make the tables unreadable and add little, we feel, to the paper. We have presented the interquartile range for all the continuous variables in the tables.
We have performed an analysis by gender and age (in table 1). We have used the chi-squared test and Wilcoxon-rank sum test to examine gender and age, respectively. We feel that these tests are robust and appropriate for the analysis. We have also presented the ages of the entire cohort, stratified by disease severity in Supplementary Figure 2. Could the reviewer be more specific about how he/she would like the data analysed and how this would enhance the paper?
Change: No change
- figure 4, the number of shared areas need to be inluded.
Response: We thank the reviewer for the suggestion. We note that the other 4 reviewers had no concerns about the presentation of figure 4. We do have the numbers of individuals who had more than one of the interventions listed in the footnote to the figure (we think this is what the Reviewer is referring to when he/she says “the number of shared areas need to be inluded”)
“There were 9 patients who received vasopressor support, RRT and mechanical ventilation; 6 patients who received vasopressor support and RRT; 5 patients who received vasopressor support and mechanical ventilation; 36 patients who received only vasopressor support and 3 patients who received only RRT
Change: No change
- there are quite a few format and spelling issues, such as there should be a blank between number and unit.
Response: We thank the reviewer for this feedback. We are happy for the production team to add a space between numbers and units if this is the Journal policy. We cannot see any spelling issues, but we would be happy to correct them if the Reviewer could highlight them for us?
Change: No change
Reviewer 4 Report
Comments and Suggestions for Authors
The article is very well written and presents very important data from a public health perspective, providing guidance for clinical suspicion and management of leptospirosis. While the data are useful in the context of Tropical Australia, they can be used as an example to compare management in other regions and replicate the clinical management of patients with leptospirosis, given the excellent clinical outcomes demonstrated.
Some clarifications or corrections are needed:
- Line 95 defines a case of leptospirosis. If the authors have the data, it would be helpful to indicate the percentage of patients diagnosed according to each of the criteria.
- In Table 1, review the numbers for serovar Australis (All 12/59, No severe disease 6/40, Severe disease 6/29).
- Table 1 indicates serovar Zanoni: 37%, Australis: 20%; line 90 indicates serovar Zanoni: 36%, Australis: 21%. Standardize these data.
- In line 406, the authors indicate the sensitivity found for PCR and serology. They should clarify which test is being compared to determine this sensitivity.
- Table 3 presents hematological values and their association with disease severity. Among other data, an association (p < 0.02) is observed with case severity and initial hemoglobin values. In the limitations section, it should be clarified that some patients may have had low values before contracting leptospirosis, for better interpretation of these data.
Author Response
The article is very well written and presents very important data from a public health perspective, providing guidance for clinical suspicion and management of leptospirosis. While the data are useful in the context of Tropical Australia, they can be used as an example to compare management in other regions and replicate the clinical management of patients with leptospirosis, given the excellent clinical outcomes demonstrated.
Some clarifications or corrections are needed.
- Line 95 defines a case of leptospirosis. If the authors have the data, it would be helpful to indicate the percentage of patients diagnosed according to each of the criteria.
Response: We thank the reviewer for raising this issue. We do list the number of patients who had each test patients who had each of the tests performed in the results (lines 184-188). Here we also describe how many of the tests were positive.
As we highlight in the discussion, in some patients the diagnosis of leptospirosis was a retrospective one. As not all patients had every test performed, as the tests were performed at different times in the patient’s illness and as the tests were performed at different sites with different turnaround times (as we highlight in lines 115-120, 372-375) it is difficult to compare the performance of the diagnostic tests in more detail.
Change: No change.
- In Table 1, review the numbers for serovar Australis (All 12/59, No severe disease 6/40, Severe disease 6/29).
Response: We thank the reviewer for noting this typographical error. This has been corrected.
Change: Correction of typographical error (Table 1)
- Table 1 indicates serovar Zanoni: 37%, Australis: 20%; line 90 indicates serovar Zanoni: 36%, Australis: 21%. Standardize these data.
Response: We thank the reviewer for noting this typographical error. This has been corrected.
Change: Correction of typographical error (Lines 189-191 and Table 1)
- In line 406, the authors indicate the sensitivity found for PCR and serology. They should clarify which test is being compared to determine this sensitivity.
Response: We thank the reviewer for highlighting this loose terminology. We have revised the text to remove the reference to sensitivity.
Change: Revision of the text
- Table 3 presents hematological values and their association with disease severity. Among other data, an association (p < 0.02) is observed with case severity and initial hemoglobin values. In the limitations section, it should be clarified that some patients may have had low values before contracting leptospirosis, for better interpretation of these data.
Response: We agree with the reviewer on this point. We have revised the limitations section accordingly (lines 385-390).
Change: Revision of the text
Reviewer 5 Report
Comments and Suggestions for Authors
An article that respects structure and methodology. It presents an important topic that has a highly relevant resonance in global public health.
From a structural point of view it is well designed, respecting the rigor of the MDPI Pathogens MDPI journal.
Author Response
Reviewer 4
An article that respects structure and methodology. It presents an important topic that has a highly relevant resonance in global public health.
From a structural point of view it is well designed, respecting the rigor of the MDPI Pathogens MDPI journal.
Response: We thank the reviewer for the time that he/she has taken to review our manuscript and are encouraged to hear that he/she found it to be well designed and rigorous. We are heartened to hear that he/she has no suggestions for the paper’s enhancement.
Change: No change
Round 2
Reviewer 1 Report
Comments and Suggestions for Authors
The term “only” introduces a subjective interpretation of the data - suggesting that the number is small or unexpected - rather than presenting the result in a neutral, descriptive manner. In scientific writing, interpretation should be reserved for the Discussion section, while the Results section should report findings factually and without value-laden language. This is not merely a matter of style, but a fundamental principle of scientific reporting. I therefore maintain that the use of “only” in the Results section is inappropriate and should be revised everywhere.
Tables: Table titles should do more than simply label the content; they should briefly describe the study population, the method or data source used, and the type of data being presented. This helps readers immediately understand the context of the table without having to search through the text. For example, instead of a vague title like "Clinical Characteristics," a more informative title would be "Clinical Characteristics of Hospitalized Leptospirosis Patients (N=111) in Queensland, Australia (study time period)." I recommend revising all table titles accordingly to improve clarity and usability.
The fact that a similar table format was used in a previous publication is not, in itself, a justification for continuing with a layout that does not follow widely accepted scientific conventions. Table formatting should prioritise clarity, consistency, and readability, and should not be based solely on past practice. Including a percentage next to every single value unnecessarily clutters the table. The standard and clearer approach is to indicate the format (e.g., N (%) or as noted in a footnote).
The term sensitivity has a specific meaning in the context of diagnostic test evaluation: it refers to the proportion of true positive cases correctly identified by the test, calculated as true positives / (true positives + false negatives). This requires a gold standard or reference to determine the true disease status. I recommend the authors carefully assess whether their use of the term follows this definition. If the appropriate data to calculate sensitivity are not available or the context does not involve diagnostic accuracy, the term should not be used, as it may lead to confusion or misinterpretation.
Author Response
- The term “only” introduces a subjective interpretation of the data - suggesting that the number is small or unexpected - rather than presenting the result in a neutral, descriptive manner. In scientific writing, interpretation should be reserved for the Discussion section, while the Results section should report findings factually and without value-laden language. This is not merely a matter of style, but a fundamental principle of scientific reporting. I therefore maintain that the use of “only” in the Results section is inappropriate and should be revised everywhere.
Response: We thank the Reviewer for the time that he/she has taken to review our revised manuscript. Our use of the word “only” is obviously distracting. We are happy to remove it from the results.
Change: Removal of the word “only” everywhere it appears it the results.
- Tables: Table titles should do more than simply label the content; they should briefly describe the study population, the method or data source used, and the type of data being presented. This helps readers immediately understand the context of the table without having to search through the text. For example, instead of a vague title like "Clinical Characteristics," a more informative title would be "Clinical Characteristics of Hospitalized Leptospirosis Patients (N=111) in Queensland, Australia (study time period)." I recommend revising all table titles accordingly to improve clarity and usability.
The fact that a similar table format was used in a previous publication is not, in itself, a justification for continuing with a layout that does not follow widely accepted scientific conventions. Table formatting should prioritise clarity, consistency, and readability, and should not be based solely on past practice. Including a percentage next to every single value unnecessarily clutters the table. The standard and clearer approach is to indicate the format (e.g., N (%) or as noted in a footnote).
Response: We thank the Reviewer for their suggestions for enhancing the presentation of the tables. We have revised all the tables with his/her suggestions in mind.
Change: Revision of the tables to address all of the Reviewer’s concerns. We have revised the titles to the format suggested. We have removed all % symbols from the tables. We have added a footnote as suggested to highlight the N (%).
- The term sensitivity has a specific meaning in the context of diagnostic test evaluation: it refers to the proportion of true positive cases correctly identified by the test, calculated as true positives / (true positives + false negatives). This requires a gold standard or reference to determine the true disease status. I recommend the authors carefully assess whether their use of the term follows this definition. If the appropriate data to calculate sensitivity are not available or the context does not involve diagnostic accuracy, the term should not be used, as it may lead to confusion or misinterpretation.
Response: We thank the reviewer for highlighting our loose terminology. We have revised the text to remove the reference to sensitivity of the tests in our cohort (lines 367-372).
Change: Revision of the text as suggested.